# Uncovering perturbations in human hematopoiesis associated with healthy aging and myeloid malignancies at single-cell resolution

**Marina Ainciburu[1,2†], Teresa Ezponda[1,2†], Nerea Berastegui[1], Ana Alfonso-Pierola[2,3], Amaia Vilas-Zornoza[1,2], Patxi San Martin-Uriz[1,2], Diego Alignani[4], Jose Lamo-Espinosa[3], Mikel San-Julian[3], Tamara Jiménez-Solas[5], Felix Lopez[5], Sandra Muntion[5,6], Fermin Sanchez-Guijo[5,6], Antonieta Molero[7], Julia Montoro[7], Guillermo Serrano[8], Aintzane Diaz-Mazkiaran[2,8], Miren Lasaga[9], David Gomez-Cabrero[9,10], Maria Diez-Campelo[5], David Valcarcel[7], Mikel Hernaez[8], Juan P Romero[1,2*‡], Felipe Prosper[1,2,3,6*‡]**

[1]Area de Hemato-Oncología, Centro de Investigación Médica Aplicada, Universidad de Navarra, Instituto de investigación sanitaria de Navarra (IDISNA), Pamplona, Spain; [2]Centro de Investigación Biomédica en Red de Cáncer, Madrid, Spain; [3]Clinica Universidad de Navarra, Pamplona, Spain; [4]Flow Cytometry Core, Universidad de Navarra, Pamplona, Spain; [5]Hospital Universitario de Salamanca, Salamanca, Spain; [6]Red de Investigación Cooperativa en Terapia Celular TerCel, ISCIII., Madrid, Spain; [7]Department of Hematology, Vall d'Hebron Hospital Universitari, Barcelona, Spain; [8]Computational Biology Program, Universidad de Navarra, Pamplona, Spain; [9]Translational Bioinformatics Unit, NavarraBiomed, Pamplona, Spain; [10]Biological & Environmental Sciences & Engineering Division, King Abdullah University of Science and Technology, Thuwal, Saudi Arabia

*For correspondence:
jromeror@unav.es (JPR);
fprosper@unav.es (FP)

†These authors contributed
equally to this work
‡These authors also contributed
equally to this work

**Abstract** Early hematopoiesis is a continuous process in which hematopoietic stem and progenitor cells (HSPCs) gradually differentiate toward specific lineages. Aging and myeloid malignant transformation are characterized by changes in the composition and regulation of HSPCs. In this study, we used single-cell RNA sequencing (scRNA-seq) to characterize an enriched population of human HSPCs obtained from young and elderly healthy individuals.

Based on their transcriptional profile, we identified changes in the proportions of progenitor compartments during aging, and differences in their functionality, as evidenced by gene set enrichment analysis. Trajectory inference revealed that altered gene expression dynamics accompanied cell differentiation, which could explain aging-associated changes in hematopoiesis. Next, we focused on key regulators of transcription by constructing gene regulatory networks (GRNs) and detected regulons that were specifically active in elderly individuals. Using previous findings in healthy cells as a reference, we analyzed scRNA-seq data obtained from patients with myelodysplastic syndrome (MDS) and detected specific alterations of the expression dynamics of genes involved in erythroid differentiation in all patients with MDS such as TRIB2. In addition, the comparison between transcriptional programs and GRNs regulating normal HSPCs and MDS HSPCs allowed identification of regulons that were specifically active in MDS cases such as SMAD1, HOXA6, POU2F2, and RUNX1 suggesting a role of these transcription factors (TFs) in the pathogenesis of the disease.

In summary, we demonstrate that the combination of single-cell technologies with computational analysis tools enable the study of a variety of cellular mechanisms involved in complex biological

systems such as early hematopoiesis and can be used to dissect perturbed differentiation trajectories associated with perturbations such as aging and malignant transformation. Furthermore, the identification of abnormal regulatory mechanisms associated with myeloid malignances could be exploited for personalized therapeutic approaches in individual patients.

## Editor's evaluation

This study generated an important single-cell transcriptome dataset using young/aged hematopoietic stem/progenitor cells obtained from normal individuals and those with MDS. The new resource provides a convincing dataset to understand a unique transcriptional landscape in elderly individuals, compared to young individuals, proving the hematopoietic aging at a transcriptome level. This manuscript will be of interest to readers in the field of hematopoiesis and associated diseases, aging, and single-cell RNA sequencing.

## Introduction

Mature blood and immune cells are generated by hematopoiesis, which is a well-characterized process that has been studied for more than a century (*Jagannathan-Bogdan and Zon, 2013*). Classically, hematopoiesis has been modeled as a stepwise differentiation process, at the core of which reside hematopoietic stem cells (HSCs), which are common precursors with self-renewal capacity. A tree-like hierarchy arises from HSCs, in which lineage commitment occurs at binary branching points, giving rise to functionally and phenotypically homogeneous progenitor populations (*Laurenti and Göttgens, 2018*; *Notta et al., 2016*). However, recent single-cell studies have questioned the validity of the classical model of hematopoiesis and have revealed heterogeneity within the HSC compartment and within progenitor populations that were previously considered homogeneous (*Haas et al., 2018*). Furthermore, it is difficult to establish boundaries between populations, which has led to the replacement of this concept by the idea of smooth transitions. Thus, early hematopoiesis is now viewed as a continuous landscape composed of undifferentiated HSPCs with a variable degree of priming toward specific lineages (lymphoid, myeloid, or erythroid) (*Velten et al., 2017*; *Karamitros et al., 2018*; *Watcham et al., 2019*; *Buenrostro et al., 2018*).

During aging, the hematopoietic system undergoes various changes. There is evidence of an aging-related increase in the relative number of HSCs, that presumably aims to compensate for a loss of repopulation ability (*Dykstra et al., 2011*; *Pang et al., 2011*). Despite this increase in the number of HSCs, the total bone marrow cellularity decreases (*Ogawa et al., 2000*). Additionally, a loss of lymphocyte production and a skewing toward myeloid differentiation have been demonstrated (*Young et al., 2016*). Moreover, recent studies have revealed an increase in the number of platelet primed HSCs (*Grover et al., 2016*). A general loss of immune function that affects both innate and adaptive immunity has also been associated with aging (*Weiskopf et al., 2009*). The molecular basis of these phenotypic changes includes an increased rate of random mutations in hematopoietic progenitors. Clonal hematopoiesis is a common trait of elderly individuals, in which blood cell subpopulations are clonally derived from a single HSC or progenitor with acquired mutations. Notably, some of these mutations are also associated with hematopoietic malignances (*Xie et al., 2014*; *Jaiswal et al., 2014*; *McKerrell et al., 2015*; *Genovese et al., 2014*). Additionally, aging is also accompanied by transcriptional dysregulation. Gene expression analyses have revealed changes in cell cycle regulators (*Kowalczyk et al., 2015*), higher expression of myeloid signatures (*Grover et al., 2016*; *Kowalczyk et al., 2015*) and genes associated with leukemia (*Rossi et al., 2005*), stronger response to inflammatory stimuli (*Martinez-Jimenez et al., 2017*; *Mann et al., 2018*), and upregulation of pathways such as nuclear factor kappa beta (NF-κB) or tumor necrosis factor alpha (TNF-α) and downregulation of DNA repair (*Chambers et al., 2007*). Lastly, epigenetic modifications that are in line with the transcriptional lesions detected have been observed in HSPCs obtained from elderly individuals (*Sun et al., 2014*).

Aging is associated with a higher risk of developing myeloid malignances (*Zeidan et al., 2019*; *Deschler and Lübbert, 2006*), suggesting a strong predisposition of hematopoietic cells from elderly individuals to lead to further alterations. Myelodysplastic syndromes (MDS) are among the main aging-related hematological disorders. MDS are characterized by ineffective hematopoiesis and

**eLife digest** Our blood contains many different types of cells; red blood cells carry oxygen through the body, platelets help to stop bleeding and a variety of white blood cells fight infections. All of these critical components come from a pool of immature cells in bone marrow, which can develop and specialise into any of these. However, as we get older, these immature cells can accumulate damage, including mutations in specific genes. This increases the risk of diseases such as myelodysplastic syndromes (MDS), a type of cancer in which the cells cannot develop and the patient does not have enough healthy mature blood cells.

The changes in gene activity in the immature cells have previously been studied using samples from young and elderly people, as well as individuals with MDS. These studies examined large numbers of cells together, revealing differences between young and elderly people, and individuals with MDS. However, this does not describe how the different types alter their behaviour.

To address this, Ainciburu, Ezponda et al. used a technique called single-cell RNA sequencing to study the gene activity in individual immature blood cells. This revealed changes associated with maturation that may account for the different combinations of cell populations in younger and older people. The results confirmed findings from previous studies and suggested new genes involved in ageing or MDS. Ainciburu, Ezponda et al. used these results to create an analytical system that highlights gene activity differences in individual MDS patients that are independent of age-related changes.

These results provide new insights that could help further research into the development of MDS and the ageing process. In addition, scientists could study other diseases using this approach of analysing individual patients' gene activity. In future, this could help to personalise clinical decisions on diagnosis and treatment.

predisposition to transformation into acute myeloid leukemia (*Shallis et al., 2018*), a highly aggressive neoplasm.

Collectively, previous evidence suggests the existence of a progressive decline in the hematopoietic system with age, which can result in disease if certain pathological events occur. HSC damage is the primary consequence of this process and its outcomes range from lineage bias to differentiation blockade and leukemic transformation (*Chung and Park, 2017*).

In this work we present a computational roadmap that can be used as a basis to fully exploit the potential of single-cell RNA sequencing (scRNA-seq) data. We applied a set of computational algorithms to assess cellular subpopulations, differentiation trajectories, and gene regulatory elements in the human hematopoietic system. This proposed methodology not only allows the characterization of healthy or control scenarios, but also the identification of specific perturbations. This is demonstrated by describing aging-dependent and pathological alterations in hematopoiesis.

## Results

### Transcriptional profiling of human young and elderly hematopoietic progenitor systems

To investigate the aging-dependent changes in the hematopoietic system, we performed scRNA-seq of bone marrow CD34$^+$ cells obtained from five young (18–20 years of age) and three elderly (>65 years of age) healthy donors (*Figure 1A*). Briefly, single-cell libraries were prepared using a Chromium controller instrument (10× Genomics) and sequenced up to an average depth of 30,000 reads per cell. A total of 34,590 and 40,641 cells were profiled from young and elderly donors, respectively. We first constructed an integrated dataset with cells from both young and elderly donors using an integration procedure implemented in the Seurat R package (*Stuart et al., 2019*). Subsequently, we subjected the integrated dataset to quality control filtering and dimensionality reduction to obtain a reference map for visualization (*Supplementary file 1*). Secondary analyses including unsupervised clustering, identification of cluster markers, trajectory inference, and gene regulatory network (GRN) reconstruction were performed independently for each group.

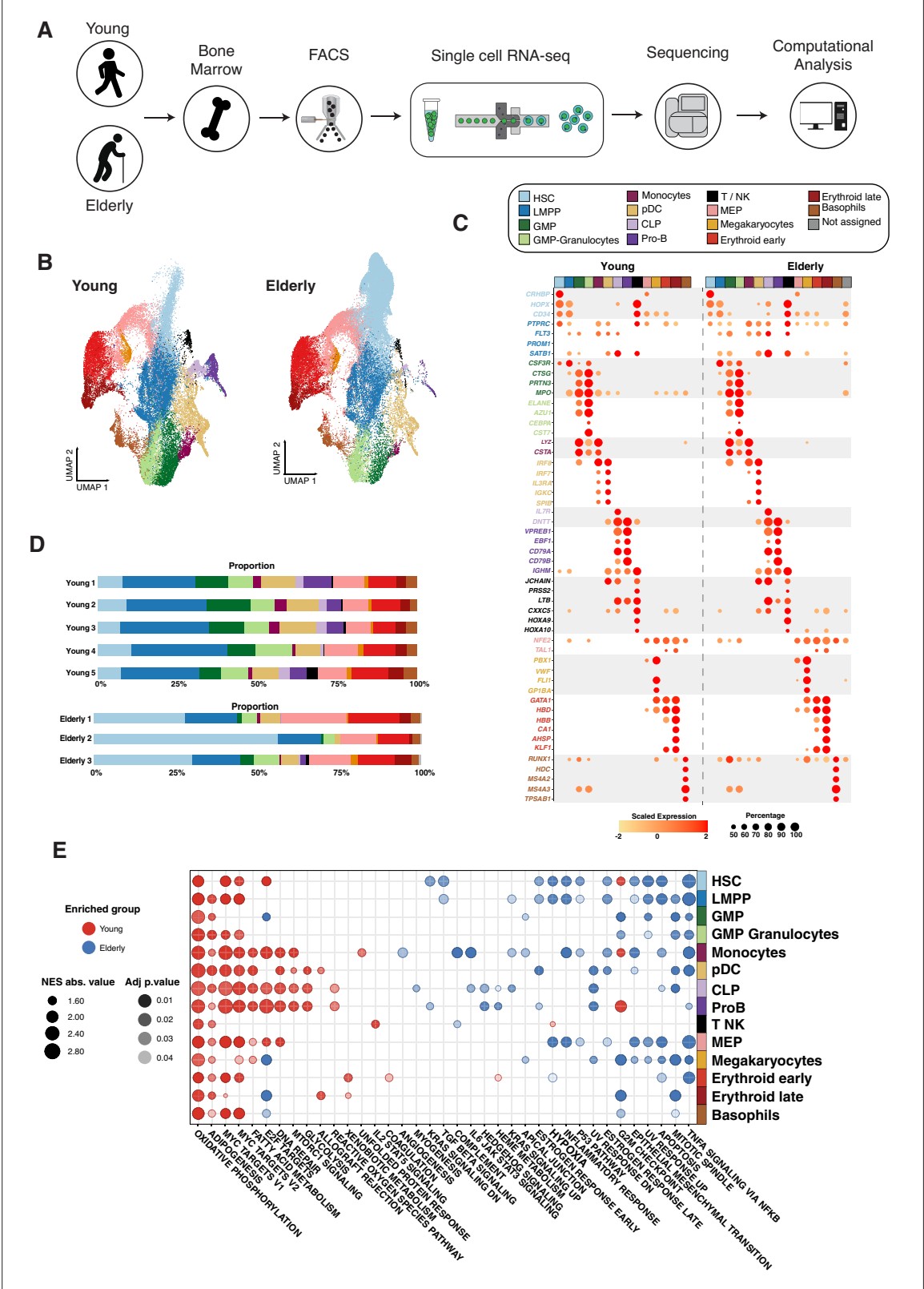

**Figure 1.** Transcriptional profiling of CD34+ cells from young and elderly healthy donors. (**A**) CD34+ cells were obtained from bone marrow aspirates of young (n=5) and elderly (n=3) donors and subjected to single-cell RNA sequencing. (**B**) UMAP plot with young cells colored according to unsupervised clustering results (left) and elderly cells labeled using an in-house cell classifier (right). (**C**) Dot plot of cluster markers (adjusted p-value <0.05) for the different cellular subpopulations identified. Dot size represents the percentage of cells that express each marker, and color represents scaled expression

*Figure 1 continued on next page*

*Figure 1 continued*

values. (**D**) Bar plots showing the proportion of cells assigned to each cellular subpopulation for each donor independently. (**E**) Dot plot of enriched terms after performing gene set enrichment analysis (GSEA) for each identified cluster. Dot color represents the enriched group, size indicates the NES absolute value, and transparency indicates the adjusted p-value.

The online version of this article includes the following figure supplement(s) for figure 1:

**Figure supplement 1.** Evaluation of GLMnet classification method.

**Figure supplement 2.** Classification of CD34+ cells in individual young and elderly donors.

**Figure supplement 3.** CD34+ progenitor proportions by flow activated cell sorting (FACS).

**Figure supplement 4.** Differentially expressed genes upon aging.

After integration, we extracted the cells obtained from young donors and subjected them to unsupervised clustering and differential expression analysis to identify cluster markers for manual cell-type annotation and labeling (*Supplementary file 2*). This analysis identified 14 cellular subpopulations comprising the landscape of HSCs, early progenitors: megakaryocyte-erythroid progenitors (MEPs), lympho-myeloid primed progenitors (LMPPs), common lymphoid progenitors (CLPs), and granulocyte-monocyte progenitors (GMPs), and cells already committed to specific lineages (pro-B cells, monocytic, erythroid, megakaryocytic, basophil and dendritic cell progenitors) (*Figure 1B*). Early progenitors were characterized by the expression of genes such as *CRHBP*, *HOPX* (initial lympho-myeloid differentiation), and *PBX1* (initial megakaryocyte and erythroid lineage). More mature progenitors presented myeloid (*MPO*, *CTSG*, *PRTN3*, *LYZ*, and *IRF8*), lymphoid (*DNTT*, *VPREB1*, and *EBF1*), erythroid (*HBD* and *CA1*), and basophil (*HDC* and *MS4A2*)-specific markers (*Figure 1C*). Once cellular subpopulations from the young donors were identified, we labeled cells of elderly individuals using a classification method based on a logistic regression with elastic-net regularization (see Materials and methods). Regularized logistic regression has been previously applied to scRNA-seq data, due to its high interpretability and good performance with sparse input (*Torang et al., 2019*; *Nguyen et al., 2018*). Briefly, we created one classifier per cell type that receives as input the normalized gene by cell expression matrix and returns a per-cell probability to be a specific cell type. Using this method we classified the integrated elderly dataset to assign cell-type probabilities (*Figure 1—figure supplement 1A*). The performance of our classifier was determined by using a publicly available dataset containing 8176 human CD34+ progenitors with known identities (*Granja et al., 2019*). This dataset was labeled using our method and Seurat (*Figure 1—figure supplement 1B*) for benchmarking purposes. We obtained the proportion of cells classified within each cell type using both approaches (*Figure 1—figure supplement 1C*). Overall, both methods provided similar results. We observed the biggest differences when assessing progenitor cell types such as HSCs, MEPs, and LMPPs. Seurat classified 38% of HSCs in the reference as MEPs, compared to 12% by GLMnet. We also observed that 83.31% of LMPPs were correctly classified by GLMnet compared to 57.91% by Seurat. The lymphoid lineage showed the biggest proportion changes; however, this is related to the low number of B cells in the reference. Based on these results, we decided to use our method in the rest of the analyses.

When we applied the GLMnet classifier to the elderly individuals, we noted a significant increase in the proportion of HSCs and a reduction in the number of both committed lymphoid and myeloid lineages (adjusted p-value <0.05, *Supplementary file 3*) as well as an increase in the proportion of erythroid-committed cells in the overall proportion, both in the integrated and individual datasets (*Figure 1D* and *Figure 1—figure supplement 2*). We used flow activated cell sorting (FACS) as an orthogonal method and observed similar results for HSCs. However, changes in the proportion of GMPs and MEPs were less obvious than in the case of the transcriptomic analysis (*Figure 1—figure supplement 3*).

Next, we checked for functional differences between the young and elderly populations using gene set enrichment analysis (GSEA) after performing differential expression per cluster and using the genes overexpressed in each condition (young and elderly) within each cell type (*Supplementary file 4*) as an input to GSEA. Cells from elderly donors exhibited a generalized enrichment of pathways activated in response to external molecules and insults, such as TNF-α, transforming growth factor beta (TGF-β), hypoxia, or inflammation. Furthermore, the p53 pathway and apoptosis programs, which are activated by stressor stimuli, were also enriched in these elderly cells (*Figure 1E*). In particular, the TGF-β response and apoptosis genes, together with genes upregulated in response to ultraviolet

radiation, were specially enriched in aging-associated HSCs. These results suggested the presence of an aging-related response to the known more inflammatory microenvironment that is present in elderly individuals (*Leimkühler and Schneider, 2019*). Shifts in the abovementioned stress-related pathways were associated with increased expression in the elderly of potential drivers of such pathways, including *JUNB*, *FOSB*, *FOS, ID2,* or *DDIT4* (*Figure 1—figure supplement 4*). Hematopoietic progenitors from young donors were characterized by an increase in metabolism, including signatures related to glycolysis, fatty acid metabolism, and oxidative phosphorylation, in components of the respiratory chain (*UQCR*, *NDUF*, and *COX*), and the glycolytic enzymes *PGK1* and *TPI1*. These findings point toward an increased metabolic activity in progenitors from young donors. We also found that cell cycle progression (*E2F* targets) and proliferation (*MYC* targets) pathways were enriched in multiple cell populations from young donors (*Figure 1E*), whereas *MYC* expression was generally downregulated in cells from elderly individuals, suggesting an aging-mediated decreased proliferative activity. DNA repair was also found to be more active in cells isolated from young individuals. This goes in agreement with the known predisposition of aging-associated progenitors to accumulate genetic lesions (*Jaiswal and Ebert, 2019*; *Mohrin et al., 2010*; *Beerman et al., 2014*).

Overall, we found that the proportion of cell types changed with age, observing a decrease in the most mature lymphoid and myelomonocytic compartments. Furthermore, GSEA revealed aging-related transcriptional alterations in all compartments, which suggests an altered biological behavior of HSPCs with age.

## Trajectory inference revealed age-specific drivers along lineages

To study the distinct differentiation trajectories observed during hematopoiesis and, specifically, alterations during aging, we subjected the single-cell expression datasets to trajectory inference using STREAM (*Chen et al., 2019*). We first inferred the differentiation trajectories of the young donors to obtain a reference that could serve as a basis for comparison. These trajectories revealed a common starting point that comprised early progenitors (HSCs, LMPPs, and MEPs). This initial root reached the branching point that divides cellular differentiation into the three main branches that resemble the main hematopoietic lineages: the myeloid, erythroid, and lymphoid lineages (*Figure 2—figure supplement 1A*). The identity of these branches was confirmed by plotting the expression profiles of previously described genes, such as *IRF8*, *GATA1,* and *EBF1* (*Figure 2—figure supplement 1B*). Then, in order to identify aging-dependent alterations, the elderly dataset was projected onto the reference trajectories using the approach described in STREAM (*Figure 2—figure supplement 1C*). We observed an increased proportion of HSCs in the root node, a reduced number of cells biased to both the myeloid and lymphoid lineages, and an increased number of progenitor cells committed to the erythroid compartment in the elderly individuals (*Figure 2—figure supplement 1D*).

Next, we aimed to identify shifts in the gene expression dynamics along differentiation trajectories. We applied the Palantir algorithm (*Setty et al., 2019*), with which we recovered a general pseudotime (*Figure 2A*) and differentiation potential (*Figure 2B*) for each cell. To measure the similarity between both methods, we performed a correlation analysis and noticed that the pseudotime values were strongly correlated ($r$=0.78) (*Figure 2—figure supplement 1E*). We observed significant differences in the distributions of differentiation potential and pseudotime, with aging-associated HSPCs accumulating greater differentiation potential and lower pseudotime (*Figure 2—figure supplement 1F*). This suggested an immature cell state, which was consistent with our previous analysis. We could also define six trajectories corresponding to six committed compartments: that is, erythroid, lymphoid, dendritic cells, monocytes, basophils, and megakaryocytes, and assigned each cell a probability of belonging to a specific branch (*Figure 2C*).

As an example of study of a particular trajectory, we focused on early monocytic differentiation, as it appeared to be impaired in the elderly donors. We plotted, in a per-cell basis, the associated pseudotime along the monocytic trajectory with the Palantir probability of reaching the final stage of the studied route (*Figure 2D*). This analysis revealed that HSCs and LMPPs from elderly donors had a higher probability of attaining the monocyte progenitor state, suggesting a stronger bias toward the monocytic compartment. However, GMPs displayed the opposite behavior which suggested that although aging-associated progenitors appeared to have a stronger initial bias toward this lineage, a large number of more advanced progenitor cells lost the capacity for monocytic differentiation. This result is in line with the aging-associated decrease in monocytic progenitors described above.

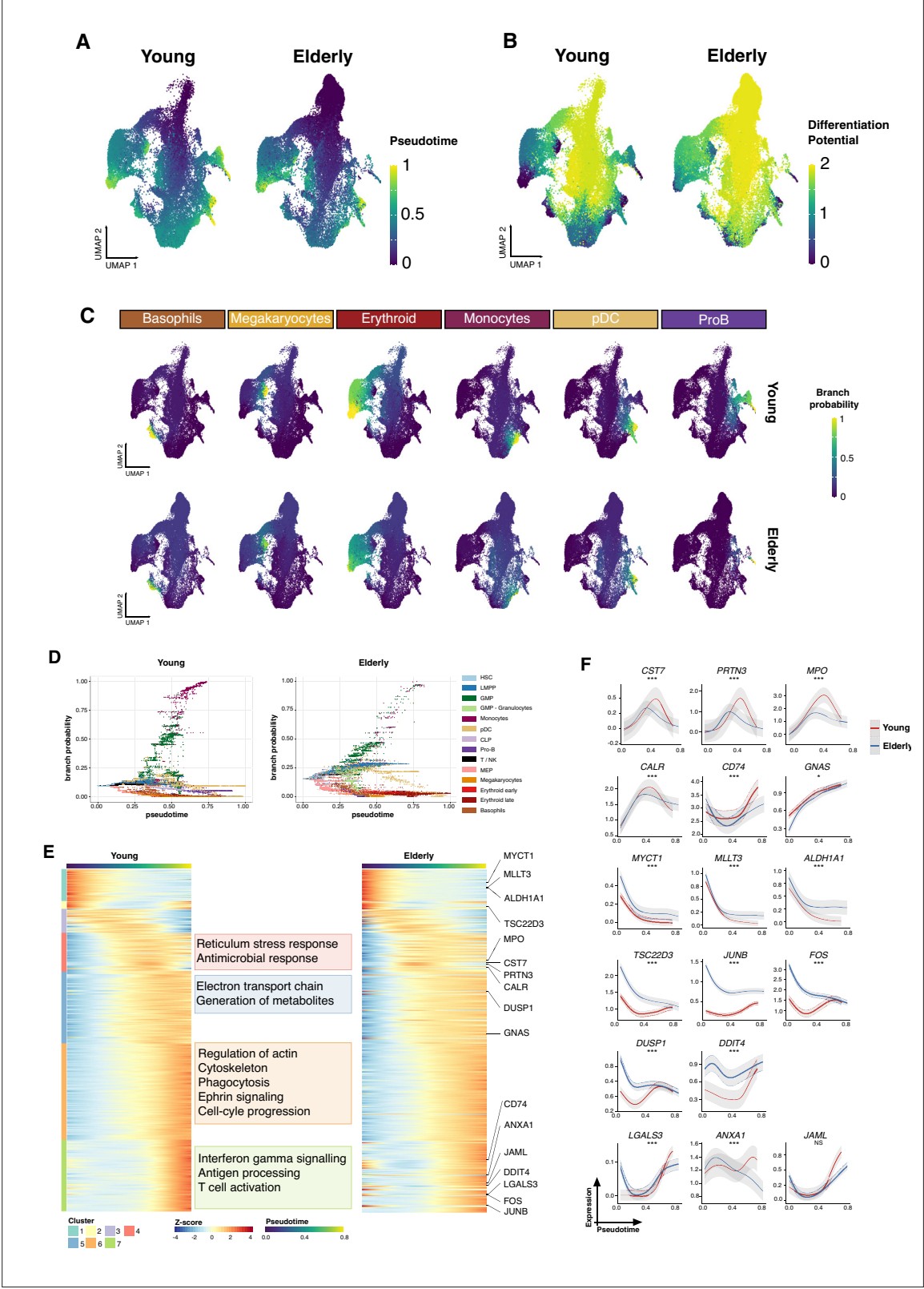

**Figure 2.** Trajectory inference of the hematopoietic lineages at single-cell resolution. (**A**) UMAP plots showing the results from applying Palantir algorithm to young and elderly cells. For both datasets, a hematopoietic stem cell (HSC) was established as initial state, based on UMAP coordinates. Final states were only indicated for the elderly dataset, as the UMAP nearest neighbors to the six young final points. Cells are colored by pseudotime and (**B**) differentiation potential. (**C**) Branch probabilities for each of the six differentiation paths retrieved. (**D**) Scatter plot of pseudotime vs. branch

*Figure 2 continued on next page*

*Figure 2 continued*

probabilities for the monocytic trajectory obtained using Palantir for young and elderly donors. Color represents the cellular subpopulation. (**E**) Heatmap of gene expression trends for dynamic genes along the monocytic trajectory in young and elderly donors. The columns are arranged according to pseudotime values, and the rows are grouped according to gene clustering results. A summary of enriched terms for the gene clusters in young donors is shown. (**F**) Expression trends in the comparison of young and elderly cells regarding the different genes involved in the monocytic trajectory (NS = not significant, *adjusted p-value <0.05, **adjusted p-value <0.01, ***adjusted p-value <0.001).

The online version of this article includes the following figure supplement(s) for figure 2:

**Figure supplement 1.** Trajectory inference with STREAM reveals the main hematopoietic differentiation branches.

To determine the transcriptional lesions that potentially alter monocytic differentiation during aging, we computed gene expression trends and clustered them based on their dynamics along the monocytic differentiation trajectory, following the Palantir pipeline. We characterized young gene clusters by over-representation analysis. Terms related to the immune system and monocytic function were enriched as the trajectory progressed, such as antimicrobial response or regulation of the actin cytoskeleton in intermediate stages, and antigen processing and presentation, or interferon gamma signaling, in gene clusters expressed near the terminal state. On computing trends for the cells of elderly donors, we observed that the overall behavior was similar. Nevertheless, genes with altered expression patterns in the elderly donors were visible (*Figure 2E*). Several myeloid differentiation markers (*CST7*, *PRTN3*, *MPO*, *CD74*, *CALR*, and *GNAS*) were expressed at lower levels across monocytic differentiation in the elderly donors, suggesting a less differentiated state. Accordingly, genes characteristic of stem or very early progenitor cells (*MYCT1*, *MLLT3*, and *ALDH1A1*) showed higher levels of expression in cells from the elderly donors across differentiation. Genes related to stress and inflammation response (*FOS*, *JUNB*, *TSC22D3*, *DUSP1*, and *DDIT4*), or monocyte chemotaxis and extravasation (*ANXA1*, *LGALS3*, and *JAML*), exhibited higher expression levels at the start of differentiation in the elderly, with several genes showing decreased expression at the terminal state (*Figure 2F*), further reinforcing the idea of a loss of capacity for monocytic differentiation in cells from elderly donors.

These analyses of the differentiation trajectory provided data regarding the genes that may be involved in the aberrant differentiation of aging-associated HSCs. However, the regulation of these transcriptional programs cannot be inferred from this analysis.

## GRNs guiding young and elderly hematopoiesis

To elucidate the regulatory mechanisms underlying healthy hematopoietic differentiation, we constructed lineage-specific GRNs using SCENIC (*Van de Sande et al., 2020*) for each of the datasets independently. Briefly, we obtained the set of activation regulons (transcription factor [TF] and their associated targets) for each of the cellular subpopulations and binarized its activity (on/off) in a per-cell basis using AUC values provided by SCENIC (through the AUCell algorithm). This approach enabled quantification of the proportion of cells that displayed an activated state for each regulon in each cluster. We then selected the top ranked regulons based on the regulon specificity score (RSS) (*Figure 3—figure supplement 1A and B*) to identify specific regulatory mechanisms per cell type. For the young donors, we obtained a series of GRNs guided by TFs known to be relevant for specific cell populations, such as *HOXA9* in early progenitor cells, or *CEBPG*, *PAX5*, and *GATA1* in the myeloid, lymphoid, and erythroid compartment, respectively (*Figure 3A*). For some regulons AUC values were highly correlated with the corresponding gene expression profiles of the guiding TF, whereas others such as CEBPG or TCF3 displayed a more specific dynamism of GRN activity that is not observed with gene expression. For each of the hematopoietic lineages, we observed sets of regulons that regulated the transcriptional state of specific cellular differentiation programs and also the presence of common targets among different TFs (*Figure 3B*). This observation indicates that a single TF might not only regulate cellular differentiation toward a specific route, but also contribute to other regulatory elements in other differentiation trajectories.

We performed the same analysis using the elderly dataset and observed that progenitor populations displayed a reduced proportion of cells with activated regulons. Specifically, HSCs showed a less active state of TFs, such as *MECOM* and *GATA3*, that were deemed specific to young donors (*Figure 3C*). In already committed cellular subpopulations, we observed similar proportions of activated cells between young and elderly donors, with the erythroid lineage showing the most similar

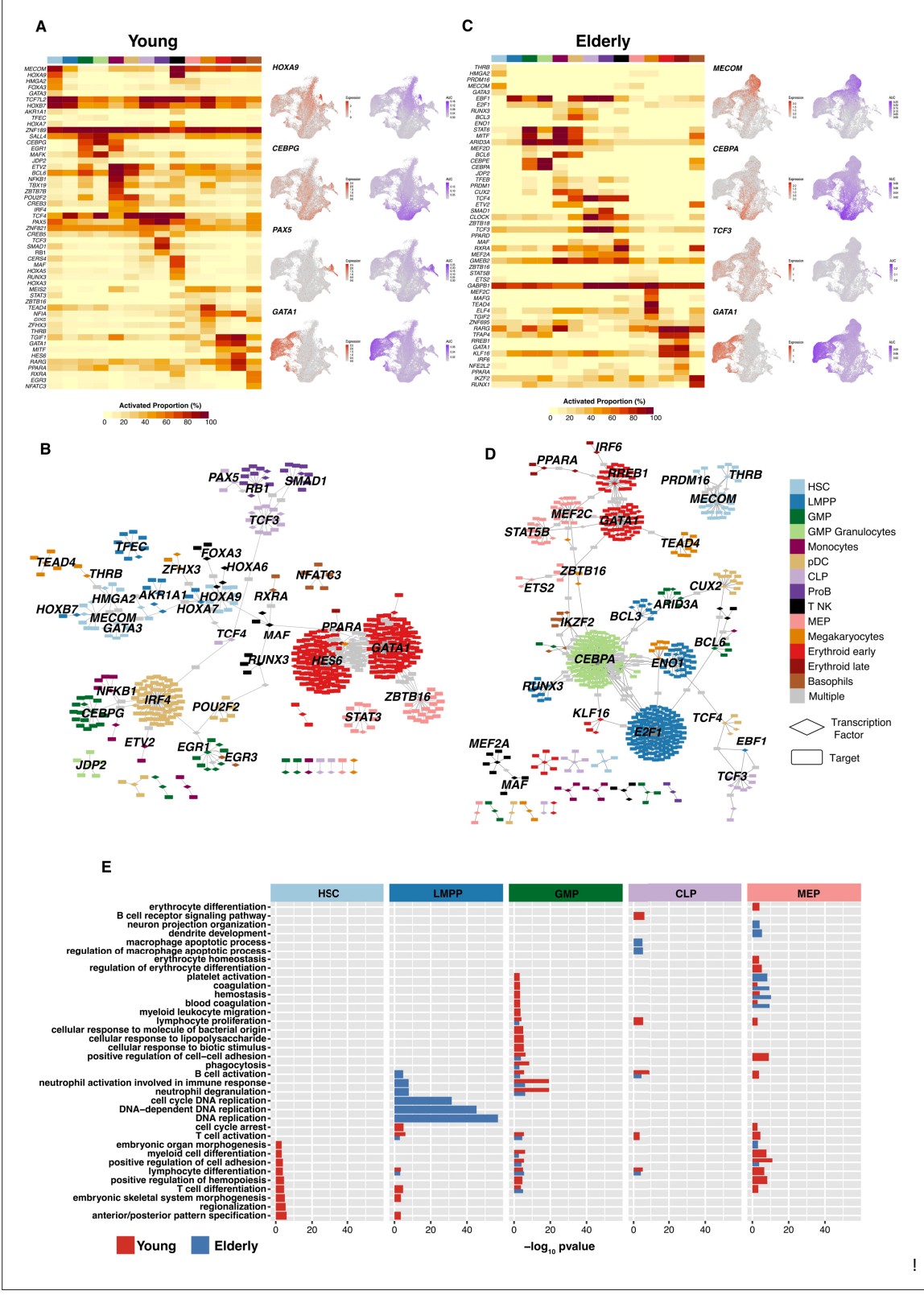

**Figure 3.** Gene regulatory network reconstruction of hematopoietic cellular populations. (**A**) (Left) Heatmap showing the proportion of cells per cluster that have an activated state for different regulons in young cells. (Right) UMAP plots with normalized expression and AUC values for specific transcription factors. (**B**) Gene regulatory network of the identified regulons for the hematopoietic system in young donors. Regulons were trimmed to include only targets with an importance score higher than the third quartile in each regulon. Node shape denotes gene-type identity, and color denotes

*Figure 3 continued on next page*

*Figure 3 continued*

cell population. Any target that can be assigned to multiple transcription factors is colored in gray. (**C**) (Left) Heatmap showing the proportion of cells per cluster that have an activated state for different regulons in elderly cells. (Right) UMAP plots with normalized expression and AUC values for specific transcription factors. (**D**) Gene regulatory network of the identified regulons for the hematopoietic system in elderly donors. Regulons were trimmed to include only the targets with an importance score higher than the third quantile in each regulon. Node shape denotes gene-type identity, and color denotes cell population. Any target that can be assigned to multiple transcription factors is colored in gray. (**E**) Bar plot with enriched gene ontology categories after over-representation analysis. Categories are grouped per cell type, and color denotes the enriched group. Bar length represents statistical significance of the enrichment, as -log10 p-value.

The online version of this article includes the following figure supplement(s) for figure 3:

**Figure supplement 1.** Extraction of cell subpopulation-specific regulons from gene regulatory networks.

profiles. However, progenitors (GMP, MEP, and CLP cells) from elderly donors showed a lower number of cells with specific regulons activated. Overall, we observed that the predicted regulatory network of elderly HSCs (*Figure 3D*) appeared as an isolated network compared to the young GRN. This finding could result in the loss of co-regulatory mechanisms in the elderly donors.

We then performed gene ontology enrichment analyses using as input the genes composing the highest ranked regulons in early progenitor compartments (HSC, LMPP, GMP, CLP, and MEP cells). HSCs from young donors displayed enriched terms related to the differentiation of hematopoietic lineages, such as myeloid cell differentiation, lymphocyte cell differentiation, and regulation of hematopoiesis, whereas HSCs from the elderly donors did not exhibit such enrichment (*Figure 3E*), suggesting a putative alteration of differentiation potential among aged HSCs. Although no increase in the percentage of LMPPs was observed in the elderly donors, this analysis indicated a clear enrichment in terms related to DNA replication in the elderly but not in the young donors, suggesting an aging-mediated alteration of the proliferative rate of this cell type. More mature progenitors also showed a youth-specific enrichment of differentiation processes such as lymphocyte differentiation, B-cell receptor signaling pathway, or T- and B-cell activation for CLPs, and erythrocyte differentiation and homeostasis for MEPs. This suggests that in aging-associated progenitor cells, relevant regulators of hematopoiesis undergo functional alterations that result in the loss of expression of genes that are required for proper hematopoietic differentiation. Collectively, these results show aging-dependent alterations in the GRNs that guide hematopoiesis, which may be associated with the diminished differentiation capabilities that early progenitors present in the elderly.

## scRNA-seq analysis of MDS specimens reveals molecular lesions affecting normal hematopoietic differentiation

To further demonstrate the potential of the used computational methods in the prediction of the mechanisms underlying aberrant hematopoietic differentiation, we explored the transcriptional alterations that distinguish normal hematopoiesis during aging from abnormal hematopoiesis associated with the development of MDS. To this end, we performed scRNA-seq from bone marrow CD34$^+$ cells obtained from four patients with MDS. In order to homogenize the group of study, and due to the great clinical and molecular heterogeneity of this disease, we focused our analyses in patients with a diagnosis of MDS with multilineage dysplasia.

We recovered 41,749 cells that passed the quality control filters (*Supplementary file 5*). All the analysis for the MDS samples was performed following the same methods used for the elderly individuals. For cell-type annotation, we predicted cell labels using our GLMnet classifier (*Figure 4A*). We noted the absence of lymphoid compartments, considerable reduction in the number of HSCs, and increase in LMPPs and GMPs subpopulations compared to healthy elderly individuals. The total percentage of erythroid compartments varied among patients, although in general it was similar to that of normal counterparts (*Figure 4B*); nevertheless, the proportion of specific erythroid populations varied, with most MDS patients showing less MEPs and more late erythroid progenitors.

Next, we carried out GSEA of genes differentially expressed between healthy elderly donors and each of the patients for the detected subpopulations. Results showed potential aberrant functionality of MDS cells. The most evident alteration was the enrichment of MDS patients in genes related to interferon alpha and gamma response, which was more evident for cases MDS1 and MDS2. This observation is in accordance with previous reports demonstrating increased inflammatory signaling in the disease (*Kim et al., 2015*; *Gañán-Gómez et al., 2015*; *Ivy and Brent Ferrell, 2018*). Interestingly,

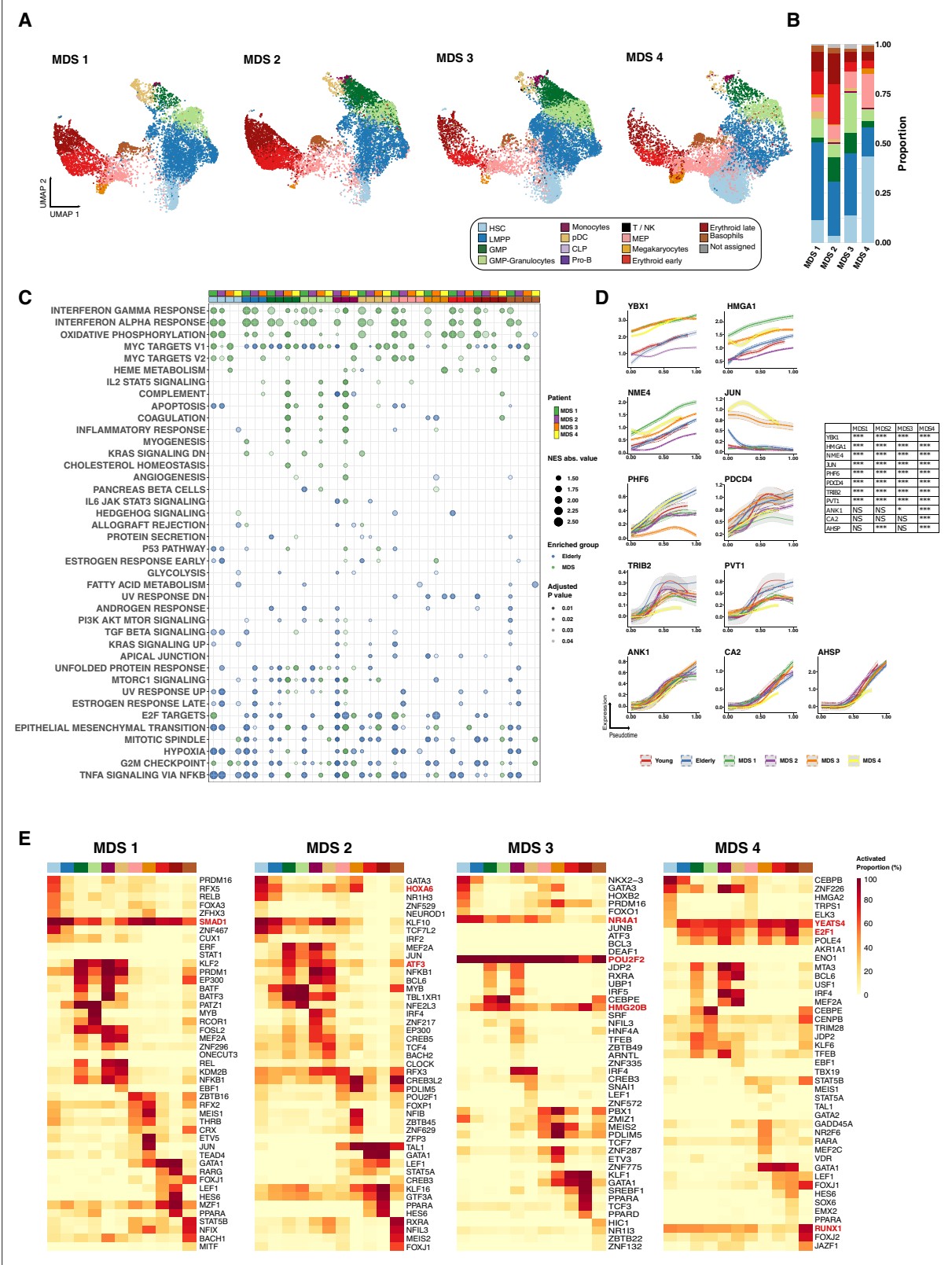

**Figure 4.** Computational analysis of pathological conditions, including myelodysplastic syndromes (MDS) and acute myeloid leukemia. (**A**) UMAP plot of CD34[+] cells from MDS (n=4). Cells are colored according to identity, as assessed using a previously described cell-type classification method. (**B**) Bar plots showing the proportion of cells assigned to each cellular subpopulation for each donor independently. Color denotes the cellular subpopulation. (**C**) Gene set enrichment analysis (GSEA) results after performing differential expression between MDS and elderly donors. Dot color represents

*Figure 4 continued on next page*

*Figure 4 continued*

enrichment direction, transparency the statistical significance, and size NES absolute value. (**D**) Expression trends in the comparison of healthy and pathological cells regarding the different genes involved in the erythroid trajectory (NS = not significant, *adjusted p-value <0.05, **adjusted p-value <0.01, ***adjusted p-value <0.001). (**E**) Heatmap showing the proportion of cells per cluster that had an activated state for different regulons in the four samples of patients with MDS among AML cells.

The online version of this article includes the following figure supplement(s) for figure 4:

**Figure supplement 1.** Computational analysis of pathological samples.

genes associated with oxidative phosphorylation, a process that is very relevant for the metabolism of several types of cancer and that is considered as an emerging target (*Ashton et al., 2018*), were more prominent in MDS. MDS patients also showed alterations of hallmarks related to cell proliferation, including E2F targets, mitotic spindle, or G2/M checkpoint, suggesting an aberrant proliferative activity of hematopoietic progenitors in these patients (*Figure 4C*). We also performed GSEA using the young donors as controls, and again found enrichment of interferon response in MDS patients. However, we observed variability in other potentially altered pathways, highlighting the importance of using age-matched controls for comparisons (*Figure 4—figure supplement 1A*).

Most MDS patients are characterized by defects in erythropoiesis, showing dysplasia and/or citopenia of this lineage. Thus, as an example of the applicability of the trajectory inference analyses to identify MDS-related transcriptional alterations, we explored gene dynamics along the HSC-erythroid branch. We reconstructed the erythroid trajectory and computed gene trends along pseudotime using Palantir (*Figure 4—figure supplement 1B*). Genes that were specifically expressed by any of the progenitors belonging to this branch (HSCs, MEPs, and erythroid progenitors) were selected and clustered according to their expression pattern (*Figure 4—figure supplement 1C*). Although we did not detect broad changes in the transcriptional profiles among the conditions, we identified specific genes participating in erythroid differentiation that displayed different dynamics from both young and elderly healthy samples (*Figure 4D*). Examples of genes with altered dynamics included increased levels of genes with a negative role in erythropoiesis such as *JUN* and *YBX1* (*Bhullar and Sollars, 2011*; *Lee et al., 2014*) and *NME4*, previously associated with poor prognosis in MDS (*Kracmarova et al., 2008*). Moreover, we also observed decreased expression across the trajectory of factors that promote erythroid differentiation, such as *TRIB2*, *PHF6*, and *PDCD4* (*Loontiens et al., 2020*; *Cho et al., 2015*) and involved in erythroleukemia: *PVT1* (*Salehi and Sharifi, 2018*). Interestingly, whereas some of the genes showed aberrant dynamics in all the patients analyzed (i.e. *TRIB2*), most of them were altered in a subset or in individual patients, reinforcing the heterogeneity of MDS at the molecular level. Despite these changes, we observed that erythroid specific markers such as *ANK1*, *CA2*, or *AHSP* followed trends similar to those of healthy HSPCs, which is in agreement with the apparently normal progression of differentiation at the progenitor stages. These results indicate that, although at very early progenitor stages some drivers of erythroid differentiation exhibit normal trends of expression, others show clear abnormalities that may manifest in more mature stages as erythroid differentiation is altered and may be responsible for the anemia and/or erythroid dysplasia that characterizes MDS patients.

Using SCENIC we analyzed the transcriptional programs regulating HSPCs and reconstructed GRNs by extracting those regulons that were specifically active in each of the cellular compartments (*Supplementary file 6*). We identified active GRNs that were guided by TFs characteristic of the different populations (*Figure 4E* and *Figure 4—figure supplement 1D*), and observed that a great number of the most prevalent regulons for each cell type were different from that of healthy donors. Interestingly, MDS cases showed regulons that were very active in most cell types, and that were not present in HSPCs from young or elderly samples, such as SMAD1 (MDS1); ATF3 and HOXA6 (MDS2); POU2F2, NR4A1, and HMG20B (MDS3); YEATS4, E2F1, and RUNX1 (MDS4). Interestingly, despite representing the same subtype, the individual MDS cases showed alterations of very specific regulons, demonstrating that the heterogeneity of the disease also takes place at the GRN level. Furthermore, some of the TFs guiding the regulons showing aberrant activity had previously been involved in the regulation of hematopoietic differentiation or the development of myeloid malignancies. For example, SMAD1 regulon was active in most cell types in patient MDS1, showing its highest activity in HSCs, LMPPs, and erythroid precursors. SMAD1 knockdown has been shown to promote erythropoiesis, suggesting that high activity of this factor may negatively impact erythroid differentiation

(*McReynolds et al., 2007*). Furthermore, SMAD1 pathway has been shown to be active in a model of persisting LSCs, suggesting that this factor may be relevant in the development of myeloid malignancies (*Lefort and Maguer-Satta, 2020*). Patient MDS2 demonstrated aberrant high activity of the regulon guided by ATF3, which has been shown to drive cell cycle progression in AML, and of that of HOXA6, a TF which potentiates hematopoietic cell differentiation and self-renewal. Patient MDS3 presented a ubiquitous activity of the regulon guided by POU2F2/OCT2, a TF overexpressed in AML, in all the subpopulations analyzed. This patient also showed high activity of NR4A1, a factor that has been shown to specify a distinct subpopulation of quiescent myeloid-biased HSCs (*Land et al., 2015*), in HSCs and LMPPs. Moreover, we also detected that HMG20B, a known repressor of erythropoiesis (*Esteghamat et al., 2011*), was prominently active in several cell types, including erythroid precursors. Finally, patient MDS4 demonstrated aberrant high activity of the regulons guided by YEATS4, which is amplified in different tumors, and E2F1, a TF whose increased activity has been previously described in MDS. These results suggested that aberrant activity of specific regulons in MDS patients could drive aberrant gene expression and ultimately promote a myelodysplastic phenotype.

Collectively, the combination of computational methods and scRNA-seq demonstrate the power of these analyses to identify novel transcriptional alterations in a personalized manner, and therefore to help uncover the molecular heterogeneity of the disease. Moreover, the approaches used do not only have the power to characterize genes with altered expression across MDS hematopoietic differentiation or any other pathological condition, but also to identify deregulated GRNs that could act as master regulators in the disease, which could be exploited therapeutically in individual patients.

## Discussion

In this study we report how the combination of computational tools can be used to generate high-resolution scRNA maps of human HSPCs and unravel changes associated with aging and disease. Although previous studies focused on other species (*Mann et al., 2018*; *Flohr Svendsen et al., 2021*) and other layers of information such as mutations (*Jaiswal and Ebert, 2019*), proteomics (*Hennrich et al., 2018*), and proteo-genomics (*Triana et al., 2021*), our study represents one of the first analyses that describes early human hematopoiesis based on its dynamic gene expression and transcriptional regulation by identifying changes that may be responsible for some of the phenotypic modifications observed during healthy aging. Applying this knowledge to HSPCs obtained from patients with MDS, we show how these analyses can help identify transcriptional alterations that play potential roles during disease development. We used MDS as a model to identify pathological alterations in the hematopoietic system, but we consider that the proposed methodology can be translated into other biological systems or pathological contexts.

Single-cell experiments are providing data at an unprecedented scale, which has allowed the identification of novel cell types and alterations that occur in biological systems. This increase in resolution has proceeded concurrently with the development of multiple computational tools that aspire to solve common problems arising with these technologies. An essential dilemma of this approach is how to establish reliable identities of the cells that will be used for subsequent analysis (*Abdelaal et al., 2019*). In this study, we generated a reference system based on the transcriptional profile of HSPCs from healthy young donors and used it to assign labels to cells from elderly healthy and pathological donors. We generated an in-house cell classifier that allowed us to reduce the effect of technical artifacts that can contribute to erroneous prediction of cell identity. This classifier was applied to cells that underwent the same sorting, library preparation, and sequencing procedures; thus, we expected to identify similar cell populations in each dataset. Additionally, when applied to external data, we found minor differences from the originally established labels, with the three lineages as well as the most immature states being precisely identified.

The results from our computational analysis go in line with previous studies, while also pointing toward genes and pathways of potential interest for further studies of hematopoiesis alterations. On one hand, we encountered an age-driven expansion of HSCs, as well as a decrease in the proportion of lymphoid precursors. Both events have been previously associated with aging (*Dykstra et al., 2011*; *Dykstra et al., 2011*; *Pang et al., 2011*; *Pang et al., 2011*; *Young et al., 2016*). Using functional analyses, we also detected an enrichment of pathways related to apoptosis and inflammatory conditions in early progenitors among elderly donors, which can be explained by the inflammatory microenvironment known to be present in elderly individuals (*Leimkühler and Schneider, 2019*).

Conversely, cells from young donors showed highly active differentiation and proliferation profiles, reinforcing the idea of the higher differentiation capacity among young individuals (*Chung and Park, 2017*). On the other hand, trajectory analyses pointed toward abnormal transcriptional dynamics across monocytic development during aging. We observed that although the most immature clusters (HSCs and LMPPs) showed an increased monocytic potential of differentiation in the elderly, more mature precursor cells presented reduced potential. This could suggest an increased priming toward the monocytic lineage of early progenitors during aging. However, alterations at later stages could result in a loss of monocytic differentiation capacity and, accordingly, a delay in the expression of genes that characterize the monocytic lineage. In agreement with our results, an aging-associated skewing toward myeloid-biased HSCs and multipotent progenitor compartments has been described in mice (*Elias et al., 2017*). By focusing on GRNs, we observed that aged progenitor cell types showed a decrease in the activity of specific regulons, a lower degree of interaction between TFs and targets, and no enrichment of pathways involved in the cellular differentiation of specific lineages. Therefore, we were able to point toward regulatory factors guiding differentiation defects in the elderly. As computational methods used to predict regulatory mechanisms in single-cell RNA-seq datasets can provide false-positive and -negative predictions (*Aibar et al., 2017*; *Pratapa et al., 2020*), other types of assays, such as ATAC-seq, or functional assays could be used as a validation strategy.

We also shed light into the molecular pathogenesis of MDS. These syndromes are characterized by a significant phenotypic and genomic heterogeneity. In that context, our methodology could have a direct clinical application by promoting the identification of patient-specific transcriptional lesions with a potential involvement in the differentiation defects. Although previous studies focused on the transcriptional lesions of MDS exist (*Hofmann et al., 2002*; *Pellagatti et al., 2006*; *Ueda et al., 2003*; *Miyazato et al., 2001*; *Im et al., 2018*; *Montalban-Bravo et al., 2020*; *Pellagatti et al., 2010*), they were performed using bulk populations of mononuclear cells and, thus, provide a limited perspective of the pathology in these patients. The detailed analysis of erythroid differentiation in our patients with MDS led to the identification of genes with altered expression dynamics, which may play a key role in promoting dyserythropoiesis. Furthermore, GRN analysis revealed key regulons, whose activity could contribute to the phenotype of these cells. Further investigations will be needed to validate the roles of these regulators in the disease.

Overall, we propose the results of this work as an in silico basis for future, in-depth studies of early hematopoietic alterations. We performed a complex analysis, taking multiple approaches, that pointed toward genes and pathways potentially involved in aging and MDS. In addition, the methodology that we developed could be applied to fully analyze datasets set in other pathological scenarios.

## Materials and methods
### Sample collection
The samples and data from the patients included in the study were provided by the Biobank of the University of Navarra and were processed according to standard operating procedures. Patients and healthy donors provided informed consent, and the study was approved by the Clinical Research Ethics Committee of the Clinica Universidad de Navarra. Bone marrow aspirates were obtained from healthy controls (young individuals [n=5], median age, 20 years, range, 19–23 years) or patients undergoing orthopedic surgery (elderly donors [n=3], median age, 72 years, range, 61–84 years). Samples from newly diagnosed patients with MDS were obtained from the Clinica Universidad de Navarra and collaborating hospitals. The patient's clinical characteristics are shown in *Supplementary file 5*.

### Fluorescence-activated cell sorting
For the CD34$^+$ cells purification, bone marrow mononuclear cells were isolated by Ficoll-Paque Plus (GE HealthCare) density gradient centrifugation and stained using CD34 (clone 8G12; BD Biosciences) CD64 (clone 10.1; BioLegend) CD19 (clone SJ25C1; BioLegend) CD10 (clone HI10A; BioLegend) CD3 (clone OKT3; BioLegend) CD36 (clone CLB-IVC7; Sanquin Plesmanlaan) CD61 (clone RUU-PL7F12; BD Biosciences) for 15 min at RT. CD34$^+$ CD64$^-$ CD19$^-$ CD10$^-$ CD3$^-$ CD36$^+$ CD61$^+$ cells were then sorted in a BD FACSAria II (BD Biosciences). Purified CD34$^+$ cells were directly used for scRNA-seq analysis.

## scRNA library preparation

The transcriptome of the bone marrow CD34$^+$ cells were examined using NEXTGEM Single Cell 3' Reagent Kits v3.1 (10× Genomics) according to the manufacturer's instructions. Between 5000 and 17,000 cells, depending on the donor, were loaded at a concentration of 700–1200 cells/μL onto a Chromium Controller instrument (10× Genomics) to generate single-cell gel bead-in-emulsions (GEMs). In this step, each cell was encapsulated with primers containing a fixed Illumina Read 1 sequence, a cell identifying 16 bp 10× barcode, a 10 bp unique molecular identifier (UMI), and a poly-dT sequence. Upon cell lysis, reverse transcription yielded full-length, barcoded cDNA, which was then released from the GEMs, amplified using polymerase chain reaction, and purified using magnetic beads (SPRIselect, Beckman Coulter). Enzymatic fragmentation and size selection were used to optimize the cDNA size prior to library construction. Fragmented cDNA was then end-repaired, A-tailed, and ligated to Illumina adaptors. A final polymerase chain reaction amplification using barcoded primers was performed for sample indexing. Library quality control and quantification were performed using a Qubit 3.0 Fluorometer (Life Technologies) and an Agilent 4200 TapeStation System (Agilent), respectively. Sequencing was performed on a NextSeq500 instrument (Illumina) (Read1: 28 cycles; Read 55 cycles; i7 index: 8 cycles) at an average depth of 30,000 reads/cell.

## scRNA preprocessing

The demultiplexing of raw base call files (BCL) was performed using the 10× software cellranger mkfastq. The generated FASTQ files were aligned to the GRCh38 version of the human genome, and count matrices were constructed using cellranger count. The default barcode filtering was performed at this step. To remove doublets, we plotted the distribution of total genes and UMIs detected per cell and established a customized superior threshold for each sample (*Supplementary file 1*). Additionally, we filtered out cells with >5% or 10% of counts landing in mitochondrial genes, as this is an indicator of dying cells. In the elderly and pathological samples, we detected a small number of cells in which <1% of the mitochondrial genes clustered together; thus, we excluded them from the analysis.

## Data integration, clustering, and visualization

Samples from the same condition (young, elderly, or MDS) were integrated using the Seurat pipeline. Gene counts were divided by the total expression per cell, multiplied by a scaling factor of 10,000, and log transformed. Normalized counts were scaled across cells. The 2000 genes with highest variance were selected using the variance stabilizing transformation method. Next, we performed integration as described previously (*Stuart et al., 2019*) using 50 dimensions. After integration, we rescaled the data, regressed the cell cycle effect, and conducted principal component analysis (PCA). Based on the visual exploration of the scree plot, we selected the appropriate number of components to continue the analysis (*Supplementary file 1*).

Next, we performed unsupervised clustering on the young integrated data, using the algorithm implemented in Seurat. We constructed a shared nearest neighbor graph based on Euclidean distances in the PCA space with the chosen dimensionality and clustered the cells using the default Louvain algorithm. We tested several resolutions and assessed the results by calculating the average silhouette for each cluster. We determined the cluster markers using the Seurat function *FindAllMarkers* with the MAST method. Next, we annotated the clusters by manually inspecting the most specific markers and searching for curated markers in the literature.

To ease visualization, we embedded the cells from the young and elderly samples under the same coordinates. Thus, we integrated the young and elderly datasets using the pipeline described above. Using the first 30 principal components, we calculated the UMAP coordinates, as implemented in Seurat, which were then used in all the figures.

## Classification model

We constructed a classification model to predict the cell identity of elderly and MDS single cells using the previously annotated young dataset as a reference. To this end, we performed logistic regression with elastic-net regularization in the glmnet R package. This approach is appropriate for sparse input data, and the elastic-net penalty allows flexible selection of variables.

We constructed an individual binary prediction model for each of the 14 identities established. To train and validate each model, we selected all the cells from the appropriate cluster and an equal

number of random cells from the remaining clusters. This set was then randomly divided in two: 75% was destined to training the model and 25% was dedicated to validation. We also performed an initial feature selection by selecting the subset of highly variable genes present in both the training-validation data (young) and the data to be classified (elderly, MDS, or AML). Initially, we trained models with different α values (1, 0.75, 0.5, 0.25, and 0.1). For each model, we used 10-fold cross-validation, and the algorithm selected the optimal value for $\lambda$. Subsequently, we selected the most appropriate α value by evaluating the performance of the models on the validation dataset. If possible, we only retained the models that used between 20 and 150 variables. Among them, we selected the model with the maximum AUC and, in case of a tie, we successively searched for the minimum false-positive rate, minimum false-negative rate, and maximum number of variables. After the optimal model was selected, we used it to classify new data as positive or negative for a particular identity.

We repeated the steps from the creation of the training and validation sets to binary prediction, 10 times and stored the resulting predictions: both the class (positive or negative) and its associated probability of being positive. Finally, we averaged the 10 resulting probabilities for each binary model and assigned the identity that corresponded to the highest probability. In cases in which none of the identities were associated with a probability of >0.5, the cell was labeled as *not assigned*.

## GSEA

We performed differential expression analysis of cells belonging to the same identity to identify differences between conditions. We applied the Seurat *FindMarkers* function to the log-normalized counts using the MAST method and the options logfc.threshold=0 and min.pct=0. In order to control for batch effect, we set patients as latent variables. In addition, we tested for all genes expressed in every sample. Only cell types with at least 25 cells per condition were used. We ranked the genes according to their average log fold change. We performed GSEA using the fgsea R package and tested for hallmark gene sets.

## Trajectory analysis using STREAM

We extracted the differentiation branches in our data using STREAM. We then used the young batch-corrected matrix to establish the reference structure, onto which we then mapped the elderly cells. Using the STREAM pipeline, we computed the 2000 most variable genes and 20 principal components. Then, we performed nonlinear dimensionality reduction using modified locally linear embedding. We projected cells onto a three-dimensional space and set the algorithm to use the 30 nearest neighbors. An elastic principal graph was constructed by computing 20 clusters and was refined using the following parameters: epg_alpha = 0.02, epg_mu = 0.05, and epg_lambda = 0.01. Subsequently, mapping was performed using the default STREAM parameters. For this step, we used the elderly batch-corrected matrix.

## Trajectory analysis using Palantir

Palantir was used to reconstruct hematopoietic lineages and recover gene expression dynamics along the lineages. This algorithm models differentiation and fate choice as a continuous probabilistic process. It orders cells along a global pseudotime, determines final states, and assigns each cell a probability to reach each terminal state.

We used the batch-corrected matrices to infer trajectories in young and elderly conditions. PCA was performed, and 30 diffusion components were calculated. For the young dataset, we selected 10 HSCs according to their position in the UMAP coordinates. We sequentially applied the Palantir algorithm using each of these cells as the initial state. Terminal states were stored for each run, and we retained those that appeared in more than five cases. If more than one terminal state corresponded to the same cell identity, we randomly selected one of them. We ran Palantir a final time adding the selected terminal states as prior information. For the elderly dataset, we selected the HSC with the highest probability, as determined by the GLMnet model, as the initial state. We then searched for cells with the highest similarity to the terminal states found in the young dataset. To this end, we calculated the nearest neighbor in the UMAP space. In the case of the MDS samples, we independently analyzed each patient. We used the normalized counts matrices, calculated 30 diffusion components, and set 10 eigen vectors to determine the multiscale space.

For each dataset, gene expression trends along pseudotime were calculated as described in Palantir using generalized additive models, with the addition of the branch probabilities as prior weights for the model.

## GRNs

Python implementation of SCENIC was used for GRN inference. All required inputs were downloaded from https://resources.aertslab.org/cistarget/ and https://pyscenic.readthedocs.io/en/latest/installation.html. Raw UMI matrices were provided as input for the pyscenic algorithm, as described in https://pyscenic.readthedocs.io/en/latest/tutorial.html. RSS were obtained using the *regulon_specificity_scores* command.

AUC matrices were imported into R and batch-corrected using the *removeBatchEffect* implemented in the limma R package. Batch-corrected AUC matrices were binarized using the BinarizeAUC function implemented in the AUCell R package. The percentage of active regulons per cell type was determined by counting the number of cells with a value of 1 in the binarized AUC matrix and dividing it by the number of cells per cell type. The top five regulons per cell type were selected according to the RSS.

Networks were drawn in Cytoscape, and only edges with an importance value larger than the third quartile (per regulon) were retained.

We performed over-representation analysis for the different sets of genes that compose the top five specific regulons for HSC, LMPP, GMP, CLP, and MEP, both in data from young and elderly donors. We tested for the gene ontology biological processes gene sets.

## Basic statistical analysis

We carried out pairwise comparison between pairs of proportions to test for significant differences in cell population proportions. We did so both among individuals from the same condition and between conditions. Two-sample Wilcoxon test was performed to test for differences in Palantir pseudotime, differentiation potential, and branch probabilities between young and elderly and per cell population. It was also used to find significant differences in gene expression trends across pseudotime between conditions. Multiple testing was addressed by adjusting p-values using the Bonferroni-Holm correction method. Results were considered significant when adjusted p-values <0.05.

## Acknowledgements

We particularly acknowledge the patients for their participation and the Biobank of the University of Navarra for its collaboration. This work was supported by the Instituto de Salud Carlos III and co-finance by FEDER funds (PI17/00701, PI19/00726, and PI20/01308), CIBERONC (CB16/12/00489 and CB16/12/00225); Gobierno de Navarra (ERAPerMed MEET-AML 0011-2750-2019-000001; AGATA 0011-1411-2020-000010/0011-1411-2020-000011 and DIANA 0011-1411-2017-000028/0011-1411-2017-000029/0011-1411-2017-000030); Fundación La Caixa (GR-NET NORMAL-HIT HR20-00871); and Cancer Research UK (C355/A26819) and FC AECC and AIRC under the Accelerator Award Program. NB was supported by a PhD fellowship from Gobierno de Navarra (0011-0537-2019-000001); MA was supported by a PhD fellowship from Ministerio de Ciencia, Innovación y Universidades (FPU18/05488); TE was supported by an Investigador AECC award from the Fundación AECC. MH was supported by H2020 Marie S Curie IF Action, European Commission, Grant Agreement No. 898356

## Additional information

### Competing interests

Juan P Romero: Employed by 10x Genomics since February 2021; this employment had no bearing on this work. The other authors declare that no competing interests exist.

## Funding

| Funder | Grant reference number | Author |
|---|---|---|
| Instituto de Salud Carlos III | | Marina Ainciburu<br>Teresa Ezponda<br>Nerea Berastegui<br>Ana Alfonso-Pierola<br>Amaia Vilas-Zornoza<br>Patxi San Martin-Uriz<br>Diego Alignani<br>Jose Lamo-Espinosa<br>Mikel San-Julian<br>Tamara Jiménez-Solas<br>Felix Lopez<br>Sandra Muntion<br>Fermin Sanchez-Guijo<br>Antonieta Molero<br>Julia Montoro<br>Guillermo Serrano<br>Aintzane Diaz-Mazkiaran<br>Miren Lasaga<br>David Gomez-Cabrero<br>Maria Diez-Campelo<br>David Valcarcel<br>Mikel Hernaez<br>Juan P Romero<br>Felipe Prosper |
| Federación Española de Enfermedades Raras | PI17/00701 | Marina Ainciburu<br>Teresa Ezponda<br>Nerea Berastegui<br>Ana Alfonso-Pierola<br>Amaia Vilas-Zornoza<br>Patxi San Martin-Uriz<br>Diego Alignani<br>Jose Lamo-Espinosa<br>Mikel San-Julian<br>Tamara Jiménez-Solas<br>Felix Lopez<br>Sandra Muntion<br>Fermin Sanchez-Guijo<br>Antonieta Molero<br>Julia Montoro<br>Guillermo Serrano<br>Aintzane Diaz-Mazkiaran<br>Miren Lasaga<br>David Gomez-Cabrero<br>Maria Diez-Campelo<br>David Valcarcel<br>Mikel Hernaez<br>Juan P Romero<br>Felipe Prosper |

| Funder | Grant reference number | Author |
| --- | --- | --- |
| Federación Española de Enfermedades Raras | PI20/01308 | Marina Ainciburu<br>Teresa Ezponda<br>Nerea Berastegui<br>Ana Alfonso-Pierola<br>Amaia Vilas-Zornoza<br>Patxi San Martin-Uriz<br>Diego Alignani<br>Jose Lamo-Espinosa<br>Mikel San-Julian<br>Tamara Jiménez-Solas<br>Felix Lopez<br>Sandra Muntion<br>Fermin Sanchez-Guijo<br>Antonieta Molero<br>Julia Montoro<br>Guillermo Serrano<br>Aintzane Diaz-Mazkiaran<br>Miren Lasaga<br>David Gomez-Cabrero<br>Maria Diez-Campelo<br>David Valcarcel<br>Mikel Hernaez<br>Juan P Romero<br>Felipe Prosper |
| Federación Española de Enfermedades Raras | PI19/00726 | Marina Ainciburu<br>Teresa Ezponda<br>Nerea Berastegui<br>Ana Alfonso-Pierola<br>Amaia Vilas-Zornoza<br>Patxi San Martin-Uriz<br>Diego Alignani<br>Jose Lamo-Espinosa<br>Mikel San-Julian<br>Tamara Jiménez-Solas<br>Felix Lopez<br>Sandra Muntion<br>Fermin Sanchez-Guijo<br>Antonieta Molero<br>Julia Montoro<br>Guillermo Serrano<br>Aintzane Diaz-Mazkiaran<br>Miren Lasaga<br>David Gomez-Cabrero<br>Maria Diez-Campelo<br>David Valcarcel<br>Mikel Hernaez<br>Juan P Romero<br>Felipe Prosper |

| Funder | Grant reference number | Author |
|---|---|---|
| Centro de Investigación Biomédica en Red de Cáncer | CB16/12/00489 | Marina Ainciburu<br>Teresa Ezponda<br>Nerea Berastegui<br>Ana Alfonso-Pierola<br>Amaia Vilas-Zornoza<br>Patxi San Martin-Uriz<br>Diego Alignani<br>Jose Lamo-Espinosa<br>Mikel San-Julian<br>Tamara Jiménez-Solas<br>Felix Lopez<br>Sandra Muntion<br>Fermin Sanchez-Guijo<br>Antonieta Molero<br>Julia Montoro<br>Guillermo Serrano<br>Aintzane Diaz-Mazkiaran<br>Miren Lasaga<br>David Gomez-Cabrero<br>Maria Diez-Campelo<br>David Valcarcel<br>Mikel Hernaez<br>Juan P Romero<br>Felipe Prosper |
| Centro de Investigación Biomédica en Red de Cáncer | CB16/12/00225 | Marina Ainciburu<br>Teresa Ezponda<br>Nerea Berastegui<br>Ana Alfonso-Pierola<br>Amaia Vilas-Zornoza<br>Patxi San Martin-Uriz<br>Diego Alignani<br>Jose Lamo-Espinosa<br>Mikel San-Julian<br>Tamara Jiménez-Solas<br>Felix Lopez<br>Sandra Muntion<br>Fermin Sanchez-Guijo<br>Antonieta Molero<br>Julia Montoro<br>Guillermo Serrano<br>Aintzane Diaz-Mazkiaran<br>Miren Lasaga<br>David Gomez-Cabrero<br>Maria Diez-Campelo<br>David Valcarcel<br>Mikel Hernaez<br>Juan P Romero<br>Felipe Prosper |

| Funder | Grant reference number | Author |
|---|---|---|
| Gobierno de Navarra | ERAPerMed MEET-AML 0011-2750-2019-000001 | Marina Ainciburu<br>Teresa Ezponda<br>Nerea Berastegui<br>Ana Alfonso-Pierola<br>Amaia Vilas-Zornoza<br>Patxi San Martin-Uriz<br>Diego Alignani<br>Jose Lamo-Espinosa<br>Mikel San-Julian<br>Tamara Jiménez-Solas<br>Felix Lopez<br>Sandra Muntion<br>Fermin Sanchez-Guijo<br>Antonieta Molero<br>Julia Montoro<br>Guillermo Serrano<br>Aintzane Diaz-Mazkiaran<br>Miren Lasaga<br>David Gomez-Cabrero<br>Maria Diez-Campelo<br>David Valcarcel<br>Mikel Hernaez<br>Juan P Romero<br>Felipe Prosper |
| Gobierno de Navarra | DIANA | Marina Ainciburu<br>Teresa Ezponda<br>Nerea Berastegui<br>Ana Alfonso-Pierola<br>Amaia Vilas-Zornoza<br>Patxi San Martin-Uriz<br>Diego Alignani<br>Jose Lamo-Espinosa<br>Mikel San-Julian<br>Tamara Jiménez-Solas<br>Felix Lopez<br>Sandra Muntion<br>Fermin Sanchez-Guijo<br>Antonieta Molero<br>Julia Montoro<br>Guillermo Serrano<br>Aintzane Diaz-Mazkiaran<br>Miren Lasaga<br>David Gomez-Cabrero<br>Maria Diez-Campelo<br>David Valcarcel<br>Mikel Hernaez<br>Juan P Romero<br>Felipe Prosper |

| Funder | Grant reference number | Author |
| --- | --- | --- |
| "la Caixa" Foundation | GR-NET NORMAL-HIT HR20-00871 | Marina Ainciburu<br>Teresa Ezponda<br>Nerea Berastegui<br>Ana Alfonso-Pierola<br>Amaia Vilas-Zornoza<br>Patxi San Martin-Uriz<br>Diego Alignani<br>Jose Lamo-Espinosa<br>Mikel San-Julian<br>Tamara Jiménez-Solas<br>Felix Lopez<br>Sandra Muntion<br>Fermin Sanchez-Guijo<br>Antonieta Molero<br>Julia Montoro<br>Guillermo Serrano<br>Aintzane Diaz-Mazkiaran<br>Miren Lasaga<br>David Gomez-Cabrero<br>Maria Diez-Campelo<br>David Valcarcel<br>Mikel Hernaez<br>Juan P Romero<br>Felipe Prosper |
| Cancer Research UK | C355/A26819 | Marina Ainciburu<br>Teresa Ezponda<br>Nerea Berastegui<br>Ana Alfonso-Pierola<br>Amaia Vilas-Zornoza<br>Patxi San Martin-Uriz<br>Diego Alignani<br>Jose Lamo-Espinosa<br>Mikel San-Julian<br>Tamara Jiménez-Solas<br>Felix Lopez<br>Sandra Muntion<br>Fermin Sanchez-Guijo<br>Antonieta Molero<br>Julia Montoro<br>Guillermo Serrano<br>Aintzane Diaz-Mazkiaran<br>Miren Lasaga<br>David Gomez-Cabrero<br>Maria Diez-Campelo<br>David Valcarcel<br>Mikel Hernaez<br>Juan P Romero<br>Felipe Prosper |

| Funder | Grant reference number | Author |
| --- | --- | --- |
| Fundación Científica Asociación Española Contra el Cáncer | Accelerator Award Program | Marina Ainciburu<br>Teresa Ezponda<br>Nerea Berastegui<br>Ana Alfonso-Pierola<br>Amaia Vilas-Zornoza<br>Patxi San Martin-Uriz<br>Diego Alignani<br>Jose Lamo-Espinosa<br>Mikel San-Julian<br>Tamara Jiménez-Solas<br>Felix Lopez<br>Sandra Muntion<br>Fermin Sanchez-Guijo<br>Antonieta Molero<br>Julia Montoro<br>Guillermo Serrano<br>Aintzane Diaz-Mazkiaran<br>Miren Lasaga<br>David Gomez-Cabrero<br>Maria Diez-Campelo<br>David Valcarcel<br>Mikel Hernaez<br>Juan P Romero<br>Felipe Prosper |
| Associazione Italiana per la Ricerca sul Cancro | Accelerator Award Program | Marina Ainciburu<br>Teresa Ezponda<br>Nerea Berastegui<br>Ana Alfonso-Pierola<br>Amaia Vilas-Zornoza<br>Patxi San Martin-Uriz<br>Diego Alignani<br>Jose Lamo-Espinosa<br>Mikel San-Julian<br>Tamara Jiménez-Solas<br>Felix Lopez<br>Sandra Muntion<br>Fermin Sanchez-Guijo<br>Antonieta Molero<br>Julia Montoro<br>Guillermo Serrano<br>Aintzane Diaz-Mazkiaran<br>Miren Lasaga<br>David Gomez-Cabrero<br>Maria Diez-Campelo<br>David Valcarcel<br>Mikel Hernaez<br>Juan P Romero<br>Felipe Prosper |
| Gobierno de Navarra | PhD fellowship 0011-0537-2019-000001 | Nerea Berastegui |
| Ministerio de Ciencia e Innovación | PhD fellowship FPU18/05488 | Marina Ainciburu |
| Fundación Científica Asociación Española Contra el Cáncer | Investigador AECC award | Teresa Ezponda |
| H2020 Marie Skłodowska-Curie Actions | Grant Agreement No. 898356 | Mikel Hernaez |
| Gobierno de Navarra | AGATA 0011-1411-2020-000010/0011-1411-2020-000011 | Marina Ainciburu |

The funders had no role in study design, data collection and interpretation, or the decision to submit the work for publication.

## Author contributions
Marina Ainciburu, Conceptualization, Writing – original draft, Writing – review and editing, Computational analysis; Teresa Ezponda, Conceptualization, Writing – original draft, Writing – review and editing; Nerea Berastegui, Resources, Data curation, Data acquisition; Ana Alfonso-Pierola, Resources, Data acquisition; Amaia Vilas-Zornoza, Resources, Data acquisition; Patxi San Martin-Uriz, Resources, Data acquisition; Diego Alignani, Resources, Data acquisition; Jose Lamo-Espinosa, Resources, Data acquisition; Mikel San-Julian, Resources, Data acquisition; Tamara Jiménez-Solas, Resources, Data acquisition; Felix Lopez, Resources, Data acquisition; Sandra Muntion, Resources, Data acquisition; Fermin Sanchez-Guijo, Resources, Data acquisition; Antonieta Molero, Resources, Data acquisition; Julia Montoro, Resources, Data acquisition; Guillermo Serrano, Computational analysis; Aintzane Diaz-Mazkiaran, Computational analysis; Miren Lasaga, Computational analysis; David Gomez-Cabrero, Computational analysis; Maria Diez-Campelo, Resources, Data acquisition; David Valcarcel, Resources, Data acquisition; Mikel Hernaez, Computational analysis; Juan P Romero, Conceptualization, Supervision, Writing – original draft, Writing – review and editing, Computational analysis; Felipe Prosper, Conceptualization, Supervision, Writing – original draft, Writing – review and editing

## Author ORCIDs
Marina Ainciburu (iD) http://orcid.org/0000-0001-6483-1901
Tamara Jiménez-Solas (iD) http://orcid.org/0000-0001-5894-2023
Juan P Romero (iD) http://orcid.org/0000-0002-0441-4791
Felipe Prosper (iD) http://orcid.org/0000-0001-6115-8790

## Ethics
The samples and data from the patients included in the study were provided by the Biobank of the University of Navarra and were processed according to standard operating procedures. Patients and healthy donors provided informed consent, together with consent for publication. The study was approved by the Clinical Research Ethics Committee of the Clinica Universidad de Navarra, following protocol # 2017.218.

## Decision letter and Author response
Decision letter https://doi.org/10.7554/eLife.79363.sa1
Author response https://doi.org/10.7554/eLife.79363.sa2

---

# Additional files

## Supplementary files
• Supplementary file 1. Parameters used for single-cell RNA sequencing (scRNA-seq) analysis.

• Supplementary file 2. Cell-type specific markers for each of the studied conditions (adjusted p-value <0.01 and logFC >0.1).

• Supplementary file 3. Cell-type proportion test between young donors, elderly donors, and conditions.

• Supplementary file 4. Differential expression analysis results between condition and per-cell subpopulation.

• Supplementary file 5. Clinical information from the donors and patients used in this study.

• Supplementary file 6. Ranking of specific regulons per cell subpopulation and condition.

• MDAR checklist

## Data availability
All the single cell RNA sequencing data is available at Gene Expression Omnibus under accession number GSE180298. The scripts needed to replicate the analysis are deposited on GitHub: https://github.com/mainciburu/scRNA-Hematopoiesis (*mainciburu, 2023*, copy archived at swh:1:rev:3e64802fb6497d396d74d9da02e9309432c8f82b).

The following dataset was generated:

| Author(s) | Year | Dataset title | Dataset URL | Database and Identifier |
|---|---|---|---|---|
| Ainciburu M, Ezponda T, Berastegui N, Alfonso-Pierola A, Vilas-Zornoza A, San Martin-Uriz P, Alignani D, Lamo de Espinosa J, San Julian M, Jimenez T, Lopez F, Muntion S, Sanchez-Guijo F, Molero A, Montoro J, Serrano G, Diaz-Mazkiaran A, Lasaga M, Gomez-Cabrero D, Diez-Campelo M, Valcarcel D, Hernaez M, Romero JP, Prosper F | 2021 | Uncovering perturbations in human hematopoiesis associated with healthy aging and myeloid malignancies at single cell resolution | https://www.ncbi.nlm.nih.gov/geo/query/acc.cgi?acc=GSE180298 | NCBI Gene Expression Omnibus, GSE180298 |

The following previously published dataset was used:

| Author(s) | Year | Dataset title | Dataset URL | Database and Identifier |
|---|---|---|---|---|
| Granja JM | 2019 | Single-cell, multi-omic analysis identifies regulatory programs in mixed phenotype acute leukemia | https://www.ncbi.nlm.nih.gov/geo/query/acc.cgi?acc=GSE139369 | NCBI Gene Expression Omnibus, GSE139369 |

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
