## [Editor Report]

This study generated an important single-cell transcriptome dataset using young/aged hematopoietic stem/progenitor cells obtained from normal individuals and those with MDS. The new resource provides a convincing dataset to understand a unique transcriptional landscape in elderly individuals, compared to young individuals, proving the hematopoietic aging at a transcriptome level. This manuscript will be of interest to readers in the field of hematopoiesis and associated diseases, aging, and single-cell RNA sequencing.

---

## [Decision Letter]

**Decision letter after peer review:**

Thank you for submitting your article "Uncovering perturbations in human hematopoiesis associated with healthy aging and myeloid malignancies at single cell resolution" for consideration by *eLife*. Your article has been reviewed by 3 peer reviewers, one of whom is a member of our Board of Reviewing Editors, and the evaluation has been overseen by Utpal Banerjee as the Senior Editor. The following individual involved in the review of your submission has agreed to reveal their identity: Jong Kyoung Kim (Reviewer #2).

Essential revisions:

1. Validate GLMnet and apply a consistent analysis platform throughout (e.g. STREAM or Palantir for pseudotime, manual annotation): All reviewers

2. Substantiate and clarify MDS analysis: (1) Same analytical strategy should be applied for young/old and MDS patients, (2) a detailed comparison between MDS-aging, (3) add clinical details: All reviewers

3. Additional supports required for the HSC population analysis, clarify young/elderly data analysis from the clonal hematopoiesis perspective: Reviewers 1 and 2

4. Consistent data description and additional validation for age-dependent cell type changes: Reviewers 1 and 3

5. Additional data/quantitation/explanation required for GRN analysis between young and old: Reviewers 2 and 3

*Reviewer #1 (Recommendations for the authors):*

1. In Figure 1b and d, elderly2 individual expresses a significantly higher HSC proportion than the other two with reductions in other populations, which can skew an entire landscape of elderly populations. Is there any other data supporting that the proportions of HSCs from elderly donors shown in Figures 1b,d are representative?

2. Regarding the above concern, what is the age distribution of elderly individuals? In a very recent study showing the clonal diversity of HSC/MPP cells in humans (for example, Mitchell et al., Nature 2022), it has been shown that the elderly over 65 yrs dramatically reduce the clonal diversity and this might be contributed to the biased HSC proportions shown in elderly2 data. Brief information, at least an age, of individuals needs to be provided as in sup table 5 for healthy donors, if possible, and discuss the extreme bias generated in elderly2 data.

3. In Figure 1e, the authors concluded that Myc is downregulated, and proliferative activity is decreased in cells of elderly individuals. However, some of the populations are rather expanded in elderly individuals; for example, the numbers of HSC or MEP are rather higher in elderly individuals. How would the authors explain such discrepancies?

4. As a nonexpert, it is not clear to me why Seurat or GLMnet-based labeling results in dissimilar proportions of cell populations. Would differing cutoffs or measurements of Seurat have given similar numbers to GLMnet? Or would it be possible to acquire numbers similar to Seurat or GLMnet with an alternative method? It would become a much-valued resource for the community if the data presented here, from young, and elderly to MDS, are analyzed in a consistent platform with a clear rationale.

5. In Figure 3e, the authors claim that HSCs from young donors show enriched terms related to differentiation of hematopoietic lineages while elderly donors do not display such an increase. However, it is not clear whether changes in the gene expression of HSCs from young donors are attributed to uniform alterations in HSC gene expressions or due to changes in the composition of HSC subsets.

1) The gene regulatory network of HSCs in elderly donors might have undergone a global change, but it is also possible that a landscape of HSC subsets (for example, long-term, short-term, or different subsets segregated by different niche interactions) could change, consequently leading to altered gene expressions.

2) Is it possible to subcategorize HSCs, for example, cells with high differentiation genes versus low, and compare them between young versus elderly?

3) Are HSCs in both young and elderly clear enough? It is possible that intermediate/committed cells, simultaneously holding stem cell characteristics, are mixed in HSC populations.

6. Even though this paper is a resource article, explanations of the MDS data are not clear enough and additional analysis may be required to better understand the disease.

1) Are the same cell types annotated when the same method used in young/elderly is applied to MDS patients?

2) It is reasonable to conclude that MDS cases show high heterogeneity of the GRN levels and each patient has specific regulons for the disease development. If so, do the four MDS patients show differential trajectories of HSC differentiation? And how are these single-cell landscapes from 4 patients associated with genetic mutations in each case (shown in sup table5)?

3) If MDS is a heterogenic disease, what would be a common idea, which can be used for future studies and therapeutics, extracted from single-cell RNA analyses?

*Reviewer #2 (Recommendations for the authors):*

I do not recommend this manuscript for publication form based on the following reasons:

1. The main claims derived from computational predictions were not well supported by the data presented and were not experimentally validated (Major points 1 to 6 of the Public Review).

2. I think if the authors address Major point 7 of the Public Review, the value of the single-cell dataset as a resource for understanding the age-associated cellular and molecular alterations during human hematopoiesis will be greatly improved. However, I agree that this would be beyond the scope of this manuscript.

If the manuscript is revised to address these concerns, I can reconsider my recommendation.

*Reviewer #3 (Recommendations for the authors):*

1. Section 'GRNs guiding young and elderly hematopoiesis' focuses on differences in transcription factor regulatory networks between young and old. Whilst an elegant analysis, it does not appear to add much more insight than the previous sections which identify expanded HSC and impaired differentiation in the elderly datasets. We would consider reducing this section and merging it with the previous one.

[Editors' note: further revisions were suggested prior to acceptance, as described below.]

Thank you for resubmitting your work entitled "Uncovering perturbations in human hematopoiesis associated with healthy aging and myeloid malignancies at single cell resolution" for further consideration by *eLife*. Your revised article has been evaluated by Utpal Banerjee (Senior Editor) and a Reviewing Editor.

The manuscript has been improved but there are some remaining issues that need to be addressed, as outlined below:

Although the authors addressed most of the concerns, some of the major comments require additional changes to warrant publication in *eLife*. Please find the concerns raised by reviewer 2, especially ones regarding the previous major comments 2 and 3.

*Reviewer #1 (Recommendations for the authors):*

The authors performed additional analyses and experiments and adequately addressed all of my major concerns.

*Reviewer #2 (Recommendations for the authors):*

The authors should specify or highlight changes in their manuscript and rebuttal. It is difficult for me to follow changes in this revised manuscript. The revised manuscript addressed some of my previous concerns but failed to address the following points:

1. Previous major comment 2 (cell-type composition changes): The newly added flow cytometry data did not support an expansion of MEPs and a reduction of GMPs in elderly individuals predicted by scRNA-seq analysis. The sentence at line 178-179 ("We used Flow Activated Cell Sorting (FACS) as an orthogonal method to support our findings (Figure 1—figure supplement 3) and observed similar results.") should be toned down accordingly.

2. Previous major comment 3 (STREAM and Palantir): I strongly disagree with the authors's opinion that mixing the results of two different methods in the same figure can be helpful for deciding which method is better suited to specific problems. Figure 2F and G can be equally well presented with pseudotime computed by STREAM as the authors showed that pseudotime values from two methods are highly correlated. To avoid any confusion and be consistent, the authors should not mix the results of two different methods in the same figure. The results generated by Palantir should be presented in a supplementary figure to demonstrate the robustness of pseudotime analysis.

3. Previous major comment 5: What does "independent network" mean?

4. Previous major comment 6: Even though this manuscript was submitted as a "Tools and Resources" article, the authors should demonstrate the robustness of their constructed GRNs. All research papers should convincingly show that the results and predictions presented in the manuscript are robust and consistent regardless of the category of the submitted manuscript. The benchmarking papers and other research papers have already shown that all methods for constructing GRNs from scRNA-seq data (including SCENIC) have an issue of false positive and negative predictions.

*Reviewer #3 (Recommendations for the authors):*

The authors comprehensively addressed the comments. I have no further concerns.

---

## [Author Response]

Essential revisions:1. Validate GLMnet and apply a consistent analysis platform throughout (e.g. STREAM or Palantir for pseudotime, manual annotation): All reviewers2. Substantiate and clarify MDS analysis: (1) Same analytical strategy should be applied for young/old and MDS patients, (2) a detailed comparison between MDS-aging, (3) add clinical details: All reviewers3. Additional supports required for the HSC population analysis, clarify young/elderly data analysis from the clonal hematopoiesis perspective: Reviewers 1 and 24. Consistent data description and additional validation for age-dependent cell type changes: Reviewers 1 and 35. Additional data/quantitation/explanation required for GRN analysis between young and old: Reviewers 2 and 3

In order to answer all the comments raised by the reviewers and the editor we have performed additional studies that can be summarized as follows:

– Used FACS data to confirm cell type proportion changes in young and elderly individuals.

– Perform and complete new analysis on young and elderly HSCs to confirm that this cell type can be classified using the same cell type markers, however the elderly cells display a more quiescent and less active cell type.

– Compared manual vs automatic cell type classification in MDS samples, to highlight how GLMnet accurately classifies cell types within different patients.

– We have updated the manuscript to clarifly the pipeline and workflow used in each of the analysis steps. We also improved data description to simplify interpretation of comparisons

Reviewer #1 (Recommendations for the authors):1. In Figure 1b and d, elderly2 individual expresses a significantly higher HSC proportion than the other two with reductions in other populations, which can skew an entire landscape of elderly populations. Is there any other data supporting that the proportions of HSCs from elderly donors shown in Figures 1b,d are representative?

We thank the reviewer for the thoughtful question. In order to support the findings from the scRNAseq analysis, we have included Flow Activated Cell Sorting (FACS) plots in Figure 1—figure supplement 3. These plots show an increase in the number of highly enriched HSCs populations (sorted as CD34+ CD38- CD45RA- CD90+ cells) in elderly individuals compared to young donors. This has also been described in the literature and we have cited relevant papers in the introduction (1,2).

2. Regarding the above concern, what is the age distribution of elderly individuals? In a very recent study showing the clonal diversity of HSC/MPP cells in humans (for example, Mitchell et al., Nature 2022), it has been shown that the elderly over 65 yrs dramatically reduce the clonal diversity and this might be contributed to the biased HSC proportions shown in elderly2 data. Brief information, at least an age, of individuals needs to be provided as in sup table 5 for healthy donors, if possible, and discuss the extreme bias generated in elderly2 data.

In order to address this comment, we have updated Supplementary File 5 to include the age of the healthy donors. Regarding the bias in the number of HSCs in one of the elderly individuals, we suspect that this just correlates with a particular case, such as the one depicted in the Figure 1—figure supplement 3. As the collection of elderly samples was difficult and dependent on availability (samples were obtained from elderly individuals undergoing hip surgery), we decided to keep all the samples to maintain the number of individuals (3).

3. In Figure 1e, the authors concluded that Myc is downregulated, and proliferative activity is decreased in cells of elderly individuals. However, some of the populations are rather expanded in elderly individuals; for example, the numbers of HSC or MEP are rather higher in elderly individuals. How would the authors explain such discrepancies?

We thank the reviewer for this comment. In Figure 1—figure supplement 4 from the manuscript, we show significant downregulation of *MYC* in several subpopulations of elderly donors, including HSCs, LMPPs, MEPs and early erythroid progenitors. This goes hand in hand with our GSEA analysis (Manuscript Figure 1e), where we find enrichment of *MYC* target gene sets (V1 and V2) for young donors. To further confirm these results, we analyzed the *MYC* regulon activity, and we calculated scores to summarize the expression of *MYC* targets, using AUCell (Author response image 1). Once again, we observe higher activity of *MYC* in young donors.

The transcriptional program regulated by *MYC* plays a complex role in the balance between cell differentiation and self renewal. Its activity has unique consequences on the different hematopoietic lineages, as well as on HSC. In this last case, its overexpression promotes proliferation and differentiation (3). Thus, dormant HSC are characterized by low levels of *MYC*, which increase in active HSCs and later progenitors (4). In agreement with this, we have found an accumulation of quiescent HSCs in elderly donors (answer to question #5, Figure 5), which has also been observed in previous studies (5,6). This could explain the higher proliferation and *MYC* activity in HSC from young donors. Something similar could occur in other early progenitor subpopulations, such as MEP, promoting the accumulation of multipotent cells at that stage.

Overall, our hypothesis regarding the increased numbers of early progenitors (HSC, MEP) in elderly individuals arises from the lack of differentiation capabilities in elderly individuals. As cells are unable to undergo the expected transition (from progenitor state to a differentiated cell), the number of these progenitor cells increases. Compared with younger individuals, we see a higher proportion of HSC, but lower values for differentiated cells in elderly donors.

**Author response image 1. sa2fig1:** MYC activity in HSPC from young and elderly donors. Violin plots showing the activity of MYC regulon obtained with SCENIC (top), and the expression of MYC target gene sets V1 (middle) and V2 (botton), summarized in a score calculated with AUCell.

4. As a nonexpert, it is not clear to me why Seurat or GLMnet-based labeling results in dissimilar proportions of cell populations. Would differing cutoffs or measurements of Seurat have given similar numbers to GLMnet? Or would it be possible to acquire numbers similar to Seurat or GLMnet with an alternative method? It would become a much-valued resource for the community if the data presented here, from young, and elderly to MDS, are analyzed in a consistent platform with a clear rationale.

The main reason the proportions differ is related to how each method works. The anchor-based approach used in Seurat identifies “anchors” or closely related cells between a given reference and a query. This is a generalized method that can be applied to transfer labels between any given reference and a dataset to be annotated.

Instead of developing a general method to classify any single cell experiment, we decided to focus on creating specific classification models for the cell types included in our integrated young donor reference. Our method is an embedded system that combines feature selection and building the classification model. The elastic-net penalization allows to select only a subset of features to avoid overfitting.

As all the samples that are analyzed in the manuscript (elderly donors and MDS patients) should include a similar subset of cell types, we decided to apply our GLMnet classifier.

The comparison between methods is described in the Figure 1—figure supplement 1 from the manuscript. We decided to use an external dataset (Granja, et. al) and assumed those cell types to be the ground truth. In the Figure 1—figure supplement 1 B and C, it can be seen that Seurat classifies a subset of HSCs as MEPs while GLMnet returns a more similar output (based on proportions) to the defined ground truth.

To further assess the raised issue, we decided to focus on HSCs that were classified as MEPs by Seurat and HSCs by GLMnet. Author response image 2 shows that the second highest score in such cells corresponds to HSCs.

**Author response image 2. sa2fig2:** Seurat classification scores. Box-plot describing the distribution of seurat scores of cells classified as MEPs by Seurat and HSCs by GLMnet.

As it has already been described, one of the biggest challenges of single cell analyses is to transform a continuum system (such as cellular differentiation) to a discrete one. We believe that the results obtained with GLMnet, are closer to the ground truth for this specific project.

5. In Figure 3e, the authors claim that HSCs from young donors show enriched terms related to differentiation of hematopoietic lineages while elderly donors do not display such an increase. However, it is not clear whether changes in the gene expression of HSCs from young donors are attributed to uniform alterations in HSC gene expressions or due to changes in the composition of HSC subsets.1) The gene regulatory network of HSCs in elderly donors might have undergone a global change, but it is also possible that a landscape of HSC subsets (for example, long-term, short-term, or different subsets segregated by different niche interactions) could change, consequently leading to altered gene expressions.2) Is it possible to subcategorize HSCs, for example, cells with high differentiation genes versus low, and compare them between young versus elderly?3) Are HSCs in both young and elderly clear enough? It is possible that intermediate/committed cells, simultaneously holding stem cell characteristics, are mixed in HSC populations.

We thank the reviewer for these questions. We have addressed all the comments related to HSCs in a similar approach, the analysis is described below:

We performed a new analysis focused on the HSC compartment, to study its heterogeneity and whether changes that take place with aging are observed throughout the whole compartment or can be associated with specific cell subsets.

First, we confirmed the suitability of the HSC classification that we established. To give additional evidence, we plotted the expression of canonical marker genes, exclusively in the HSC compartment, separated by individual donors. We found that every donor expressed the HSC markers, whereas the majority of them did not express markers of committed cell populations from any of the hematopoietic lineages (Author response image 3).

**Author response image 3. sa2fig3:** Expression of marker genes in the HSC compartment. Dot plot depicting the normalized scaled expression of canonical marker genes by HSC of the 5 young and 3 elderly healthy donors. Marker genes are colored by the cell population they characterize. Dot color represents expression levels, and dot size represents the percentage of cells that express a gene.

Next, we chose an unsupervised approach to sub-cluster HSCs, allowing the main sources of variability among the cells to drive the process. We created two new datasets containing only HSCs, one for young donors and another one for elderly donors. We independently reintegrated each of them with Seurat, to remove donor-associated batch effects, and performed unsupervised clustering.

For the HSC coming from young donors, we chose a clustering resolution of 0.3, resulting in 4 clusters (Author response image 4) homogeneously distributed across patients (Author response image 4). Differential expression analysis returned few significant cluster specific markers (from 2 to 50), pointing towards the existence of high similarity between clusters. In the case of the HSC coming from elderly donors, we chose a resolution of 0.2, resulting in 6 clusters, three of them significantly smaller than the rest (Author response image 4). Every cluster was present in the 3 donors, though we found more heterogeneity in the distribution among patients than in the young HSC (Author response image 4). Differential expression analysis yielded >100 markers for clusters 3, 4 and 5 and <35 for clusters 0, 1 and 2.

Then, we explored whether any correspondence existed between young and elderly clusters, aiming to know if HSCs from both young and elderly cluster together due to similar biological reasons. We compared common marker genes for both sets of clusters and found the greatest overlap between the following pairs: cluster 0 from young and cluster 1 from elderly, cluster 2 from young and cluster 0 from elderly, cluster 3 from young and cluster 3 from elderly. We found that CDK6, an essential gene for HSC exit from quiescence, was a common marker for cluster 0 in young and cluster 1 in elderly (Author response image 4 top left), suggesting differences in HSC activation states among clusters. To further explore this, we scored cells for curated gene signatures characterizing quiescence (7) and proliferation (8). We observed that quiescence scores are higher in cluster 2 from young donors and cluster 0 from elderly donors compared to the rest (Author response image 4 middle left). In line with this, proliferation is especially high on a small subset of HSC in clusters 0 and 1 from young donors, a subset of HSC in cluster 1 from elderly and cluster 4 from elderly (Author response image 4 bottom left). When we assessed the cell cycle phase HSCs are in, following the Seurat pipeline, we found the great majority of them to be in G1, as expected. Cells with high proliferation scores correlate with S phase and, finally, cluster 4 from elderly donors was found to be specific to cells in G2/M phase (data not shown). Therefore, quiescence and proliferation appear to be the greatest effects driving HSC clustering.

In order to study differences in lineage priming, we also scored cells for gene signatures associated with HSC, GMP and MEP (9). We observed that HSC scores positively correlate with quiescence scores (Author response image 4 top right). On the contrary, GMP signature appears to be more active in cells with less quiescent state, and cluster 5 from elderly scored especially high for myeloid progenitors signature (Author response image 4 middle right). MEP genes did not show any cluster-related pattern of activation (Author response image 4 bottom right).

To sum up, we have found that the main driver for HSCs variability is their state of quiescence vs proliferation. Elderly donors present higher numbers of dormant HSCs, which could indeed be the cause for the lower expression of proliferation and metabolism related genes in this compartment. On the other hand, differences in lineage priming among HSCs, and age-associated changes in priming are not clear on this data.

**Author response image 4. sa2fig4:** HSC sub clustering. (A) UMAP visualization of HSC from young (left) and elderly (right) donors subjected to re-integration and unsupervised clustering. Cells are colored by clusters. (B) Bar plot showing the proportion of cells from each donor assigned to the different clusters. (C) UMAP plots for young (left) and elderly (right) HSC colored by the normalized expression of CDK6 (top left) and by the summarized expression of multiple gene signatures, quantified as scores calculated using the software AUCell.

We then focused on the HSC-specific regulons that resulted from our GRN analysis (Figure 3 from the manuscript). We aimed to explore whether differences between young and elderly donors in transcriptional programs activity are homogeneously distributed among HSC (Author response image 5). We assessed activity scores (AUC) across the HSC sub-clusters we generated with this analysis. We did not observe great differences in activity across clusters. For some transcriptional programs, such as PRDM16 or GATA3, we see slightly increased activity in the clusters corresponding to the most quiescent HSCs (cluster 2 in young and 0 in elderly), but this trend stays the same for both young and elderly.

Overall, we observe that the identified HSC population in both elderly and young donors have a common set of markers, showing subpopulations that seem to be driven by quiescence and proliferation (with higher quiescence scores in elderly). Further inspection of GRNs in these HSC clusters revealed no differences in regulation programs between conditions, suggesting that the observed patterns across conditions are equivalent.

**Author response image 5. sa2fig5:** Activity of HSC-specific regulons. Violin plots showing activity scores for the top 5 HSC-specific regulons generated by SCENIC in HSC from both young (left) and elderly (right) donors, separated by sub-clusters. Color indicates sub-cluster.

6. Even though this paper is a resource article, explanations of the MDS data are not clear enough and additional analysis may be required to better understand the disease.1) Are the same cell types annotated when the same method used in young/elderly is applied to MDS patients?2) It is reasonable to conclude that MDS cases show high heterogeneity of the GRN levels and each patient has specific regulons for the disease development. If so, do the four MDS patients show differential trajectories of HSC differentiation? And how are these single-cell landscapes from 4 patients associated with genetic mutations in each case (shown in sup table5)?3) If MDS is a heterogenic disease, what would be a common idea, which can be used for future studies and therapeutics, extracted from single-cell RNA analyses?

We thank the reviewer for this set of questions concerning the analysis of MDS samples.

Regarding the first point, cells in the 4 MDS datasets were annotated with the same method used for elderly: cell identities found in the healthy young HSPC pool, together with their gene expression profile, were used as reference to predict cell identities in the MDS datasets. To do so, we used the GLMnet-based method we developed. Cells in the MDS datasets were assigned the identity with highest probability among the ones calculated by the GLMnet binary classification models. If a minimum probability is not met for any identity, then a cell is classified as “not assigned” (see methods section for more details). This strategy was chosen as we expected the MDS samples to be composed by the same cell populations as the ones we identified in the healthy young samples. All of them underwent the same sample processing, FACS sorting, library preparation and sequencing processes. In addition, clinical data indicated that blast presence in the bone marrow of these patients was very low: between 0 and 4% (supp table 5). Therefore, this classification method assumes that the great majority of the cells belong to the established set of identities. At the same time, it allows for changes in the proportions of cell populations, as well as not to assign a label to cells that do not resemble any of the possible cell identities.

During this revision, we have performed additional validations for the classification, including clustering and manual annotation of the MDS samples. With this method, we have not found new cell types that were not present in our reference samples (detailed answer to Reviewer 2, first comment).

We have updated the manuscript to reflect the comment raised by the reviewer by adding the following phrase (Page 7, line 15): “All the analysis for the MDS samples was performed following the same methods used for the elderly individuals. For cell type annotation, we predicted cell labels using our GLMnet classifier”. We have emphasized that the same classification strategy used for elderly samples was used to classify MDS cells.

The second point made by the reviewer is of high interest. Exploring the effects of mutations in the transcriptional profile of patients could give key insight into the disease and contribute to the development of targeted treatments. In this study, however, we consider we have not enough samples to raise conclusions around this issue. Three out of four patients have mutations in genes of the spliceosome (*ZRSR2* in MDS1 and *U2AF1* in MDS2 and MDS3). MDS2 and MDS3 also share a mutation in the histone regulator *ASXL1*. However, MDS4 only has a record of a *TET2* mutation at a very low VAF (2.6%). The sample size is not big enough to associate any of the alterations we observe to specific mutations.

Instead, we chose this set of patients because they all share MDS with multilineage dysplasia, resulting in more than one myeloid lineage affected by the disease. All of them have a deficiency of erythrocytes and/or platelets. This is the reason why we focussed on the megakaryocyte-erythroid differentiation trajectory. We could consistently reconstruct this path in the 4 MDS datasets, and calculate the gene expression trends among which we looked for alterations.

As mentioned by the reviewer, MDS is a heterogeneous disease that can display specific alterations per patient. As a resource article, we intend to highlight the use of single-cell RNA sequencing as a viable tool to analyze pathological cases on a case by case basis rather than in a collective approach. Any pathological sample can be used to follow our proposed approach and identify case-specific alterations.

Also, we have identified a set of common alterations shared between our pathological donors. Among these alterations, we describe the widespread enrichment of interferon α and γ response genes in MDS samples. This could point towards deregulated interactions with the microenvironment, where higher expression of interferon genes have been found (10), or towards an altered immune system. We also observed genes with altered expression trends along erythroid differentiation in multiple MDS samples, such as TRIB2 and PHF6.

Reviewer #3 (Recommendations for the authors):1. Section 'GRNs guiding young and elderly hematopoiesis' focuses on differences in transcription factor regulatory networks between young and old. Whilst an elegant analysis, it does not appear to add much more insight than the previous sections which identify expanded HSC and impaired differentiation in the elderly datasets. We would consider reducing this section and merging it with the previous one.

We thank the reviewer for the comment regarding the GRN section of the manuscript. We consider that having an independent section for GRN analysis allows readers to better understand the corresponding section. As we use specific algorithms for the prediction or regulatory features, which are different from the trajectory analysis one, this layout avoids confusion between what can be obtained with each tool.

[Editors' note: further revisions were suggested prior to acceptance, as described below.]

Reviewer #2 (Recommendations for the authors):The authors should specify or highlight changes in their manuscript and rebuttal. It is difficult for me to follow changes in this revised manuscript. The revised manuscript addressed some of my previous concerns but failed to address the following points:1. Previous major comment 2 (cell-type composition changes): The newly added flow cytometry data did not support an expansion of MEPs and a reduction of GMPs in elderly individuals predicted by scRNA-seq analysis. The sentence at line 178-179 ("We used Flow Activated Cell Sorting (FACS) as an orthogonal method to support our findings (Figure 1—figure supplement 3) and observed similar results.") should be toned down accordingly.

We thank the reviewer for the comments. We have updated the manuscript and update the corresponding sentence (Page 4. line 27-29) as follows:

“We used Flow Activated Cell Sorting (FACS) as an orthogonal method and observed similar results for HSCs. However changes in the proportion of GMPs and MEPs were less obvious than in the case of the transcriptomic analysis (Figure 1—figure supplement 3)”

2. Previous major comment 3 (STREAM and Palantir): I strongly disagree with the authors's opinion that mixing the results of two different methods in the same figure can be helpful for deciding which method is better suited to specific problems. Figure 2F and G can be equally well presented with pseudotime computed by STREAM as the authors showed that pseudotime values from two methods are highly correlated. To avoid any confusion and be consistent, the authors should not mix the results of two different methods in the same figure. The results generated by Palantir should be presented in a supplementary figure to demonstrate the robustness of pseudotime analysis.

We thank the reviewer for the comments. We have updated Figure 2 to include all results obtained with Palantir and Figure 2—figure supplement with STREAM results. Also, we included the following sentence to emphasize that the results with both methods are strongly correlated (Page 5. line 24-26):

“To measure the similarity between both methods, we performed a correlation analysis and noticed that the pseudotime values were strongly correlated (r = 0.78) (Figure 2—figure supplement 1e).”

3. Previous major comment 5: What does "independent network" mean?

We thank the reviewer for the comment. We refer to the HSC GRN in elderly individuals as independent due to the lack of connection with other identified regulons in different cellular subpopulations. In contrast, the young HSC GRN shows a connection with the rest of elements in the network.

We have updated the corresponding section (Page 6. line 40) to better reflect the findings:

“Overall, we observed that the predicted regulatory network of elderly HSCs (Figure 3d) appeared as an isolated network compared to the young GRN.”

4. Previous major comment 6: Even though this manuscript was submitted as a "Tools and Resources" article, the authors should demonstrate the robustness of their constructed GRNs. All research papers should convincingly show that the results and predictions presented in the manuscript are robust and consistent regardless of the category of the submitted manuscript. The benchmarking papers and other research papers have already shown that all methods for constructing GRNs from scRNA-seq data (including SCENIC) have an issue of false positive and negative predictions.

We thank the reviewer for the comments and certainly agree with him that predicted GRN may need functional validation to determine their true biological value. In that sense, we have included a phrase in the discussion (Page 9. line 42-44) to reflect the fact that computational methods used to predict regulatory mechanisms can show false positive/negative results and that other strategies could be used as a validation strategy.

“As computational methods used to predict regulatory mechanisms in single cell RNA-seq datasets can provide false positive and negative predictions^(58,59)^, other types of assays, such as ATAC-seq, or functional assays could be used as a validation strategy”

References

1. Dykstra B, Olthof S, Schreuder J, Ritsema M, de Haan G. Clonal analysis reveals multiple functional defects of aged murine hematopoietic stem cells. J Exp Med. 2011 Dec 19;208(13):2691–703.

2. Pang WW, Price EA, Sahoo D, Beerman I, Maloney WJ, Rossi DJ, et al. Human bone marrow hematopoietic stem cells are increased in frequency and myeloid-biased with age. Proc Natl Acad Sci USA. 2011 Dec 13;108(50):20012–7.

3. Delgado MD, León J. Myc roles in hematopoiesis and leukemia. Genes Cancer. 2010 Jun;1(6):605–16.

4. Cabezas-Wallscheid N, Buettner F, Sommerkamp P, Klimmeck D, Ladel L, Thalheimer FB, et al. Vitamin A-Retinoic Acid Signaling Regulates Hematopoietic Stem Cell Dormancy. Cell. 2017 May 18;169(5):807-823.e19.

5. Hérault L, Poplineau M, Mazuel A, Platet N, Remy É, Duprez E. Single-cell RNA-seq reveals a concomitant delay in differentiation and cell cycle of aged hematopoietic stem cells. BMC Biol. 2021 Feb 1;19(1):19.

6. Kowalczyk MS, Tirosh I, Heckl D, Rao TN, Dixit A, Haas BJ, et al. Single-cell RNA-seq reveals changes in cell cycle and differentiation programs upon aging of hematopoietic stem cells. Genome Res. 2015 Dec;25(12):1860–72.

7. Graham SM, Vass JK, Holyoake TL, Graham GJ. Transcriptional analysis of quiescent and proliferating CD34+ human hemopoietic cells from normal and chronic myeloid leukemia sources. Stem Cells. 2007 Dec;25(12):3111–20.

8. Venezia TA, Merchant AA, Ramos CA, Whitehouse NL, Young AS, Shaw CA, et al. Molecular signatures of proliferation and quiescence in hematopoietic stem cells. PLoS Biol. 2004 Oct;2(10):e301.

9. Laurenti E, Doulatov S, Zandi S, Plumb I, Chen J, April C, et al. The transcriptional architecture of early human hematopoiesis identifies multilevel control of lymphoid commitment. Nat Immunol. 2013 Jul;14(7):756–63.

10. Kim M, Hwang S, Park K, Kim SY, Lee YK, Lee DS. Increased expression of interferon signaling genes in the bone marrow microenvironment of myelodysplastic syndromes. PLoS ONE. 2015 Mar 24;10(3):e0120602.

11. Aibar S, González-Blas CB, Moerman T, Huynh-Thu VA, Imrichova H, Hulselmans G, et al. SCENIC: single-cell regulatory network inference and clustering. Nat Methods. 2017 Nov;14(11):1083–6.

12. Pratapa A, Jalihal AP, Law JN, Bharadwaj A, Murali TM. Benchmarking algorithms for gene regulatory network inference from single-cell transcriptomic data. Nat Methods. 2020 Feb;17(2):147–54.

13. Kuranda K, Vargaftig J, de la Rochere P, Dosquet C, Charron D, Bardin F, et al. Age-related changes in human hematopoietic stem/progenitor cells. Aging Cell. 2011 Jun;10(3):542–6.

14. Rossi DJ, Bryder D, Zahn JM, Ahlenius H, Sonu R, Wagers AJ, et al. Cell intrinsic alterations underlie hematopoietic stem cell aging. Proc Natl Acad Sci USA. 2005 Jun 28;102(26):9194–9.

15. Dinauer MC, Newburger PE, Borregaard N. Phagocyte System and Disorders of Granulopoiesis and Granulocyte Function. Nathan and Oski’s Hematology and Oncology of Infancy and Childhood [Internet]. 8th ed. 2015 [cited 2022 Oct 11]. p. 773–847. Available from: https://www.clinicalkey.com/#!/content/book/3-s2.0-B978145575414400022X

16. Domínguez Conde C, Xu C, Jarvis LB, Rainbow DB, Wells SB, Gomes T, et al. Cross-tissue immune cell analysis reveals tissue-specific features in humans. Science. 2022 May 13;376(6594):eabl5197.

17. Bapat A, Schippel N, Shi X, Jasbi P, Gu H, Kala M, et al. Hypoxia promotes erythroid differentiation through the development of progenitors and proerythroblasts. Exp Hematol. 2021 May;97:32-46.e35.